# Separating mitochondrial protein assembly and endoplasmic reticulum tethering by selective coupling of Mdm10

Lars Ellenrieder[1,2], Łukasz Opaliński[1,†], Lars Becker[3], Vivien Krüger[3], Oliver Mirus[4], Sebastian P. Straub[1,2], Katharina Ebell[3], Nadine Flinner[4,†], Sebastian B. Stiller[1], Bernard Guiard[5], Chris Meisinger[1,6], Nils Wiedemann[1,6], Enrico Schleiff[4,7], Richard Wagner[3,8], Nikolaus Pfanner[1,6] & Thomas Becker[1,6]

The endoplasmic reticulum–mitochondria encounter structure (ERMES) connects the mitochondrial outer membrane with the ER. Multiple functions have been linked to ERMES, including maintenance of mitochondrial morphology, protein assembly and phospholipid homeostasis. Since the mitochondrial distribution and morphology protein Mdm10 is present in both ERMES and the mitochondrial sorting and assembly machinery (SAM), it is unknown how the ERMES functions are connected on a molecular level. Here we report that conserved surface areas on opposite sides of the Mdm10 β-barrel interact with SAM and ERMES, respectively. We generated point mutants to separate protein assembly (SAM) from morphology and phospholipid homeostasis (ERMES). Our study reveals that the β-barrel channel of Mdm10 serves different functions. Mdm10 promotes the biogenesis of α-helical and β-barrel proteins at SAM and functions as integral membrane anchor of ERMES, demonstrating that SAM-mediated protein assembly is distinct from ER-mitochondria contact sites.

[1] Institute of Biochemistry and Molecular Biology, Centre for Biochemistry and Molecular Cell Research, Faculty of Medicine, University of Freiburg, Freiburg D-79104, Germany. [2] Faculty of Biology, University of Freiburg, Freiburg D-79104, Germany. [3] Division of Biophysics, School of Biology/Chemistry, University of Osnabrück, Osnabrück D-49034, Germany. [4] Molecular Cell Biology of Plants, University of Frankfurt, Frankfurt D-60438, Germany. [5] Centre de Génétique Moléculaire, Centre National de la Recherche Scientifique, Gif-sur-Yvette 91190, France. [6] BIOSS Centre for Biological Signalling Studies, University of Freiburg, Freiburg D-79104, Germany. [7] Buchmann Institute of Molecular Life Sciences, Cluster of Excellence Macromolecular Complexes, University of Frankfurt, Frankfurt D-60438, Germany. [8] Life Sciences & Chemistry, Focus Area Health, Jacobs University Bremen, Bremen D-28759, Germany. † Present addresses: Faculty of Biotechnology, Department of Protein Engineering, University of Wroclaw, Wroclaw 50383, Poland (Ł.O.); Frankfurt Institute for Advanced Studies, University of Frankfurt, Frankfurt D-60438, Germany (N.F.). Correspondence and requests for materials should be addressed to N.P. (email: nikolaus.pfanner@biochemie.uni-freiburg.de) or to T.B. (email: thomas.becker@biochemie.uni-freiburg.de).

Mitochondria were originally considered to function as semi-autonomous organelles that produce ATP, yet are largely independent of the rest of the cell. This view has radically changed since it was found that mitochondria are deeply integrated into central cellular functions, from numerous metabolic pathways to protein and lipid biogenesis, signalling processes, quality control and apoptosis[1–4]. Recent studies led to the identification of contact sites between mitochondria and other cell organelles, providing strong evidence for the extensive integration of mitochondria into cellular homeostasis and organization[5–11].

The endoplasmic reticulum (ER)–mitochondria encounter structure (ERMES) forms a stable bridge between the ER membrane and the mitochondrial outer membrane[5]. The ERMES complex consists of four core components. The maintenance of mitochondrial morphology (Mmm1) protein is anchored in the ER membrane. The mitochondrial distribution and morphology proteins Mdm10 and Mdm34 (Mmm2) were reported to be integrated into the outer membrane of mitochondria, whereas Mdm12 is a peripheral membrane protein[5,12–14]. The stable assembly of the four components into the ERMES complex connects both organelles, however, the molecular architecture of ERMES is not fully understood. A fifth subunit, the GTPase Gem1, associates with ERMES, but is present in substoichiometric amounts and is not required for formation of the ERMES complex[15–17]. Multiple mitochondrial functions have been linked to ERMES, including biogenesis and assembly of outer membrane proteins, lipid homeostasis, membrane dynamics, mitophagy and maintenance of mitochondrial morphology[5,11,18–24]. A molecular assignment of ERMES functions has been difficult and thus different views exist which functions are directly connected to ERMES and which are indirect effects. In case of mitochondrial inheritance, it was reported that inheritance defects of ERMES mutants were indirectly caused by defects of mitochondrial morphology[17].

Whereas three ERMES core components, Mmm1, Mdm12 and Mdm34, are selectively located in the ERMES complex, the fourth one, Mdm10, has a dual localization. Mdm10 is part of both the ERMES complex and the sorting and assembly machinery (SAM) of the mitochondrial outer membrane[18,19,21,25,26]. The SAM complex is responsible for the membrane insertion of two types of newly synthesized mitochondrial outer membrane proteins, β-barrel proteins and some α-helical proteins. SAM is thus required for the biogenesis and assembly of the main translocase of the outer membrane (TOM) that consists of both β-barrel and α-helical proteins[18,26–33]. Mutants of Mdm10 disturb major ERMES-connected functions, including mitochondrial morphology, lipid homeostasis and protein assembly, yet it is open which molecular functions are performed by ERMES-bound Mdm10 and which ones by SAM-bound Mdm10. Remarkably, yeast cells lacking either Mmm1, Mdm12 or Mdm34 display alterations of the mitochondrial tubular network, of phospholipid profiles and of outer membrane protein assembly like *mdm10Δ* cells[5,13,19,25,34,35], although these components do not interact with SAM. Two explanations are conceivable to explain the mutant phenotypes: (i) either the association of Mdm10 with the SAM complex is functionally not relevant and thus *mdm10Δ* cells just display ERMES defects; or (ii) Mdm10 and the other ERMES core components are so closely linked that the deletion of entire components will impact on Mdm10 functions. Indeed, the Mdm10 molecules present in SAM or ERMES do not form strictly separate pools, but Mdm10 can shuttle between both complexes, supporting the second view[19,31]. Tom7, which has a dual localization at TOM and Mdm10, binds to SAM-free Mdm10 and thus favours a release of Mdm10 from SAM and its association with ERMES[21,32,36].

It is unknown how Mdm10 is recruited to ERMES and/or SAM. A functional dissection of Mdm10 and its binding partners will represent a key step towards assigning ERMES- and SAM-specific functions. In this study, we perform a systematic structure–function analysis of Mdm10 and its interaction with partners. We report that conserved surface areas on opposite sides of the Mdm10 β-barrel are crucial for the recruitment of Mdm10 to either ERMES or SAM. Assembly of mitochondrial outer membrane proteins is specifically linked to SAM-bound Mdm10, whereas the maintenance of mitochondrial morphology and lipid homeostasis is linked to ERMES-bound Mdm10. Our findings reveal that the β-barrel of Mdm10 plays different roles. It forms a channel for the SAM-mediated insertion of Tom22 into the outer membrane and serves as integral membrane anchor of ERMES at mitochondria, thus promoting its functions in membrane morphology and lipid transfer.

## Results

**SAM and ERMES bind to opposite sides of Mdm10.** Mdm10 belongs to the VDAC/Tom40 superfamily of eukaryotic β-barrel proteins[37,38]. The members of the superfamily consist of 19 antiparallel β-strands with a similar fold, connected by loops of variable length[37–39]. To define SAM- and ERMES-specific functions of Mdm10, we asked if Mdm10 contains different binding sites for its partner proteins. To identify potential binding sites, we analysed conserved regions of Mdm10 based on the observation that evolutionarily conserved (identical or similar) amino-acid residues, in particular hydrophobic and aromatic residues, are often enriched in protein–protein interfaces[40]. Since deletion of various loops of Mdm10 neither affected cell growth nor the interaction of Mdm10 with ERMES or SAM[38], we focused on the membrane-integrated β-barrel domain. Mdm10 contains two conserved regions on opposite sides of its β-barrel (Fig. 1a)[38]: a surface groove formed between β-strands 4 and 5 is flanked by an invariant glycine residue (G144) and two adjacent aromatic residues of β-strand 3 (conserved Y73 and Y75), whereas a strip of several conserved aromatic residues is located on the other side (including Y296 and F298 on β-strand 14 and Y301 on the subsequent loop). To study the functions of the conserved regions, we replaced aromatic residues by alanine and the glycine residue by leucine in yeast. The resulting mutant cells were impaired in growth at elevated temperature, in particular on non-fermentable medium when a high activity of mitochondria is required (Fig. 1b). To study the interaction of Mdm10 with its partners, mitochondria were lysed with the non-ionic detergent digitonin, and Mdm10-containing complexes were purified by co-immunoprecipitation. Strikingly, the mutant mitochondria showed selective differences in the interaction of Mdm10 with SAM, ERMES and Tom7. The co-purification of Sam35 was strongly reduced in the Mdm10$^{Y73,75A}$ mitochondria, whereas the co-purification of ERMES subunits (Mmm1 and Mdm12) was unaffected (Fig. 1c, lane 8). In contrast, Mdm10$^{Y296A,F298A}$ mitochondria were impaired in the interaction of Mmm1 and Mdm12, but not of Sam35, with Mdm10 (Fig. 1c, lane 10). Moreover, Mdm10$^{G144L}$ mitochondria were impaired in the co-purification of Tom7, but not of Sam35 or ERMES subunits (Fig. 1c, lane 9). We conclude that different surface areas of the Mdm10 β-barrel are crucial for the interaction with SAM and ERMES.

To exclude indirect effects of the *mdm10* site-specific mutants, we performed a number of control experiments. (i) The steady-state levels of various proteins, including SAM and ERMES subunits, were not or only mildly affected in the mutant

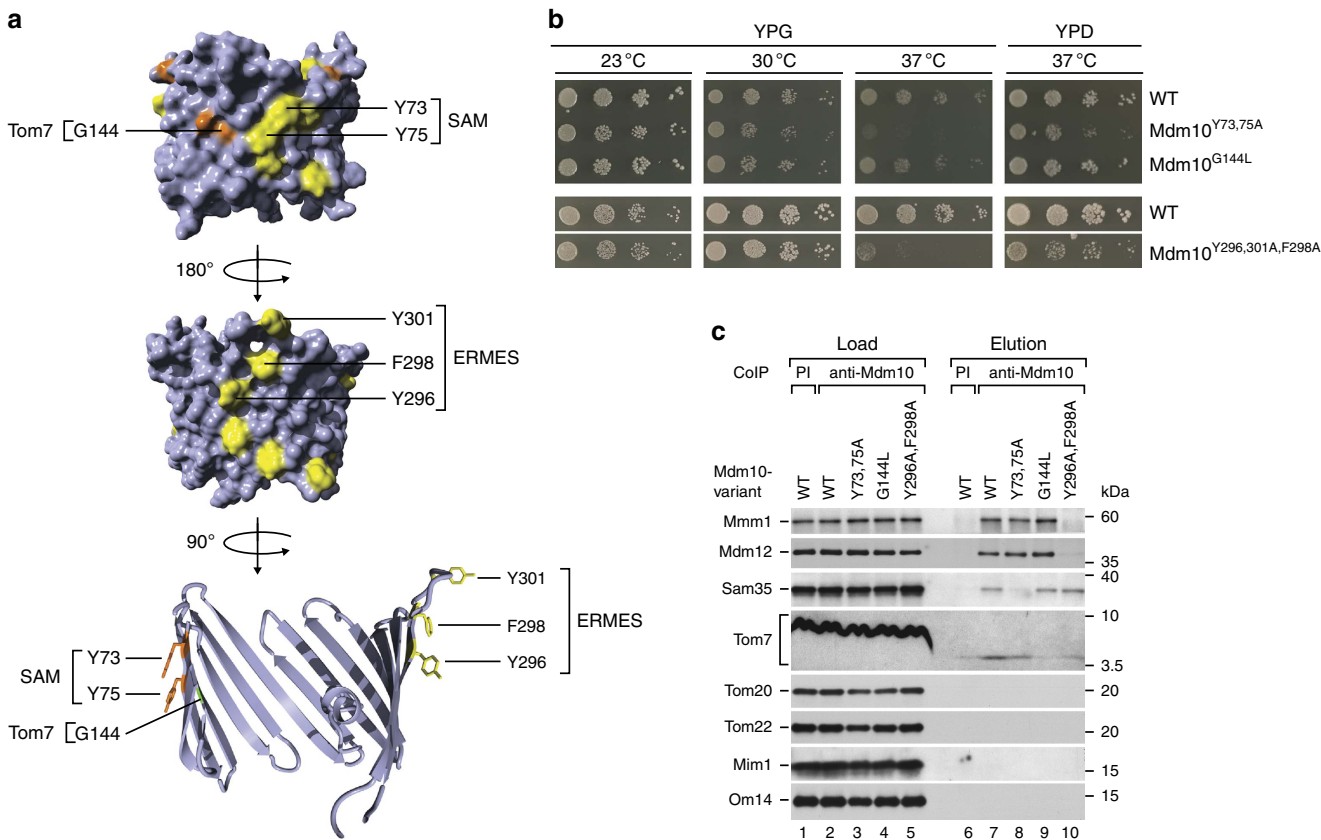

**Figure 1 | Identification of protein interaction sites on the β-barrel surface of Mdm10. (a)** Homology model of the β-barrel domain of Mdm10 (ref. 38). Long hydrophilic loops are shortened. Conserved amino-acid residues and a conserved hydrophobic groove on one side of the β-barrel (yellow (Phe, Trp and Tyr) and orange (Ala/Gly); top panel) and a strip of conserved aromatic residues on the other side of the barrel (yellow; middle panel) are highlighted. Amino-acid residues that were replaced by mutational analysis in this study are labelled. Bottom, the β-barrel of Mdm10 is depicted as cut open model in a ribbon representation. **(b)** Serial dilutions of yeast strains expressing wild-type (WT) or mutant forms of *MDM10* were grown on agar plates containing either glycerol (YPG) or glucose (YPD) as carbon source at different temperatures. **(c)** Mitochondria isolated from cells expressing WT or mutant forms of *MDM10* were solubilized with digitonin and subjected to co-immunoprecipitation (CoIP) using antibodies raised against Mdm10. Samples were analysed by SDS–polyacrylamide gel electrophoresis and immunodetection of the indicated proteins. Load 2%, elution 100%. PI, pre-immune serum.

mitochondria (Supplementary Figs 1a and 2a). (ii) Cells with a deletion of *MDM10* frequently show a loss of mtDNA[25], leading to indirect effects on mitochondrial structure and function. The *mdm10* site-specific mutants were able to grow on non-fermentable medium (Fig. 1b) and the oxidative phosphorylation complexes II, III, IV and V, analysed by blue native electrophoresis, were indistinguishable from that of wild-type mitochondria (Supplementary Figs 1b and 2b), demonstrating that the mitochondrial genome was functional. (iii) To study if the *mdm10* site-specific mutants affected the interactions of the three SAM core components Sam35, Sam37 and Sam50, we expressed Sam50 with a protein A tag. Sam35 and Sam37 were efficiently co-purified with Sam50 in all mutant mitochondria like in wild-type mitochondria (Fig. 2a), demonstrating that the SAM$_{core}$ complex remained intact. The co-purification of Mdm10 with Sam50 was selectively disturbed in the Mdm10$^{Y73,75A}$ mutant, but remained unaffected in the other Mdm10 mutant strains (Fig. 2a). To directly analyse the SAM–Mdm10 complex, we performed affinity purification via tagged Sam50 or tagged Mdm10, and analysed the elution samples by blue native electrophoresis. The SAM–Mdm10 complex was stable in Mdm10$^{Y296A,F298A}$ and Mdm10$^{G144L}$ mitochondria, and selectively compromised in Mdm10$^{Y73,Y75A}$ mitochondria (Fig. 2b), indicating that Y73/Y75 were required for the association of Mdm10 with the SAM$_{core}$ complex. (iv) The TOM–SAM supercomplex, which is transiently

formed between TOM and SAM$_{core}$ for the initial transfer of β-barrel precursors[41], was analysed by the co-purification of TOM subunits with Sam50. The *mdm10* mutants did not disturb formation of the supercomplex (Fig. 2a) in agreement with the observation that Mdm10 is not part of the TOM–SAM supercomplex[41]. (v) Since site-specific mutants of the strip of aromatic residues of Mdm10 disturbed its interaction with ERMES (Fig. 1c), we asked if the mutants destabilized the ERMES complex by probing the interactions of its subunits. We expressed protein A-tagged Mmm1 (ref. 19) in double and triple mutants of the aromatic strip, Mdm10$^{Y296A,F298A}$ and Mdm10$^{Y296,301A,F298A}$. Co-purification of Mdm10 with tagged Mmm1 was diminished in the Mdm10$^{Y296A,F298A}$ double mutant and strongly impaired in the Mdm10$^{Y296,301A,F298A}$ triple mutant, whereas the association of the other core components Mdm12 and Mdm34 with Mmm1 was not or only moderately affected (Fig. 2c and Supplementary Fig. 2c). Mutants of the aromatic strip of Mdm10 thus selectively disturb the interaction of Mdm10 with ERMES without disrupting the interaction of the other ERMES core components.

We conclude that Mdm10 contains separate binding sites for SAM and ERMES, located on opposite surfaces of the β-barrel (Fig. 2d). The site-specific *mdm10* mutants selectively affect the interaction with either SAM or ERMES, and thus represent a system to experimentally dissect SAM- and ERMES-specific functions.

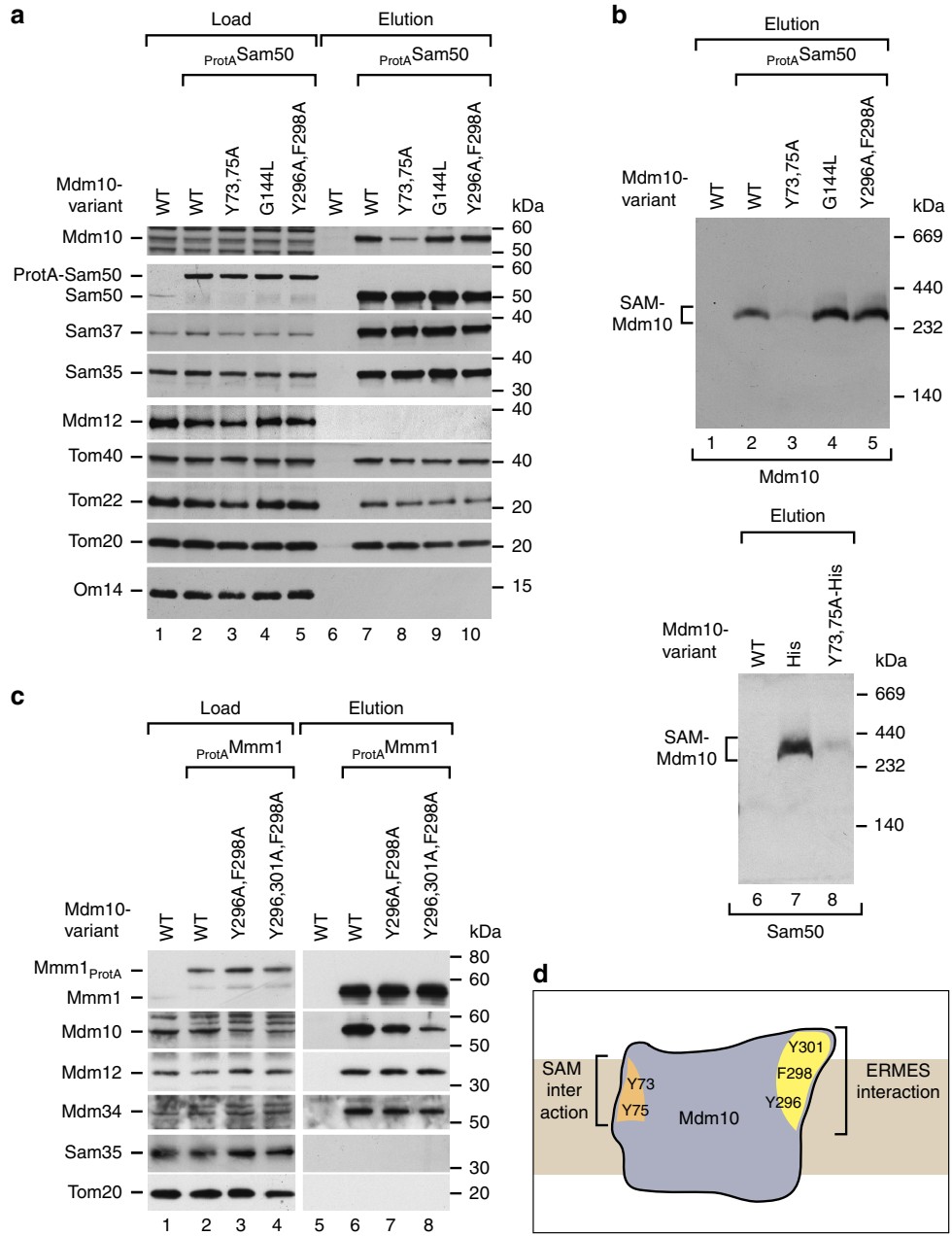

**Figure 2 | SAM and ERMES bind to different sides of the Mdm10 β-barrel.** (**a**) Whole-cell extracts of wild-type (WT) and ProtASam50 cells expressing the indicated *mdm10* mutant forms were solubilized with digitonin and subjected to affinity purification using IgG antibodies. Samples were analysed by SDS–polyacrylamide gel electrophoresis (SDS–PAGE) and immunodetection. Load 2%, elution 100%. (**b**) Upper panel: whole-cell extracts of WT and ProtASam50 cells expressing the indicated *mdm10* mutant forms were solubilized with digitonin and subjected to affinity purification. Elution samples were analysed by blue native electrophoresis and immunodetection with anti-Mdm10 antibodies. Lower panel: mitochondria isolated from WT, Mdm10His and Mdm10Y73,75A His cells were solubilized with digitonin and subjected to affinity purification using Ni-NTA agarose. Protein complexes of the elution fractions were separated by blue native electrophoresis and analysed by immunodetection with anti-Sam50 antibodies. (**c**) Whole-cell extracts of WT and ProtAMmm1 cells expressing the indicated *mdm10* mutant forms were obtained by cryo-grinding, solubilized with digitonin and subjected to affinity purification using IgG antibodies. Samples were analysed by SDS–PAGE and immunodetection with the indicated antisera. Load 2%, elution 100%. (**d**) The hypothetical model indicates that Mdm10 interacts with SAM and ERMES via different regions on the outside of its β-barrel domain (orange and yellow patches on Mdm10).

**ERMES-bound Mdm10 is linked to mitochondrial morphology.**
Deletion of *MDM10* leads to severe alterations of the normal tubular mitochondrial morphology of yeast cells[12,18,23,35,36]. We asked which population of Mdm10 was crucial for maintaining the tubular mitochondrial network. Mitochondria were stained with the fluorescence dye 3,3′-dihexyloxacarbocyanine iodide (DiOC6). The mitochondrial network was normally formed in

Mdm10Y73,75A yeast, which are impaired in the Mdm10–SAM interaction, as well as in Mdm10G144L yeast, which are impaired in the Mdm10–Tom7 interaction (Fig. 3a). However, in Mdm10Y296,301A,F298A yeast, which are defective in the Mdm10–ERMES interaction, the tubular mitochondrial network was largely destroyed and a clustering of mitochondria was observed (Fig. 3a). Quantification confirmed the strong alteration

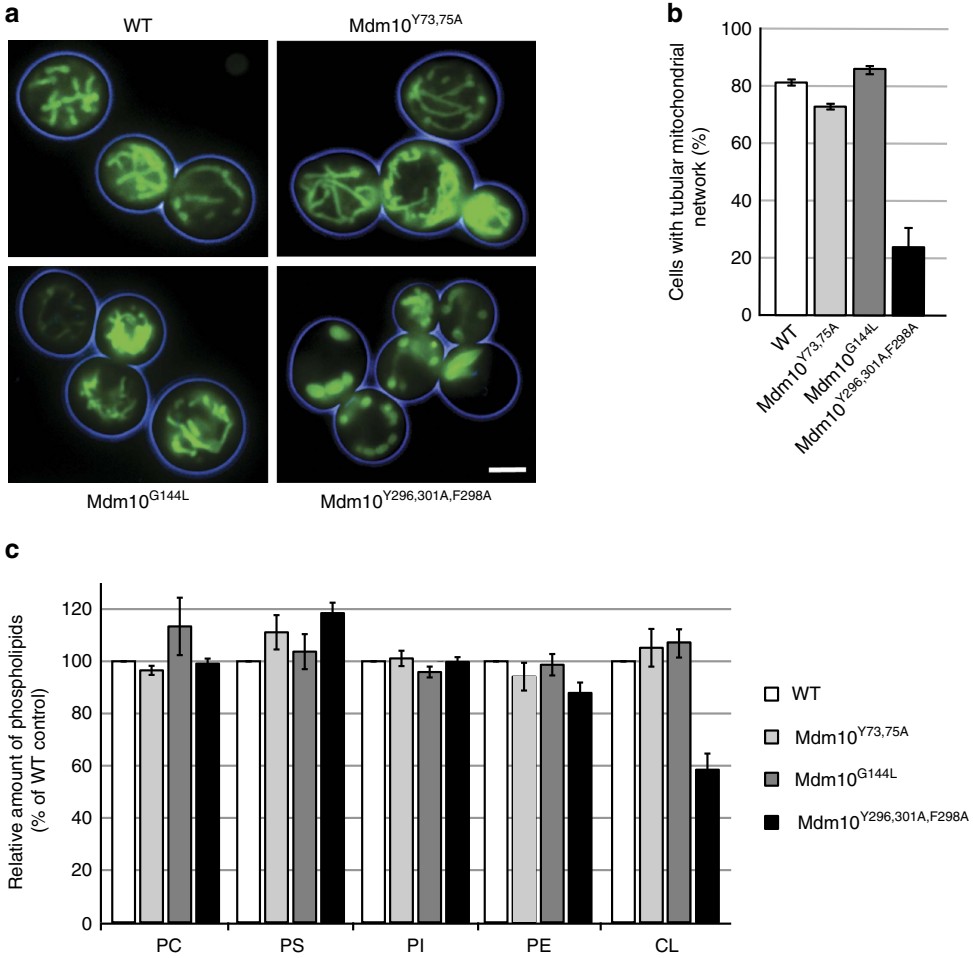

**Figure 3 | Mdm10 at the ERMES complex is crucial for maintaining mitochondrial morphology and cardiolipin levels.** (**a**) The mitochondrial morphology of yeast strains expressing wild-type (WT) or mutant forms of *MDM10* was visualized by $DiOC_6$ staining and fluorescence microscopy. Scale bar, 5 μm. (**b**) Quantification of mitochondrial morphology based on fluorescence microscopy and staining with $DiOC_6$. Relative amounts of cells with tubular network are depicted from three independent experiments with at least 300 cells analysed per strain, shown as mean ± s.e.m. (**c**) Yeast cells expressing WT or mutant forms of *MDM10* were labelled with [$^{33}$P]orthophosphate for 90 min at 37 °C. After isolation of mitochondria, phospholipids were extracted and separated by thin-layer chromatography. Radiolabelled phospholipids were visualized by autoradiography. The amount of each major phospholipid is shown in relation to its WT level (set to 100%). Data are presented as mean ± s.e.m. ($n = 3$).

of the mitochondrial morphology in Mdm10$^{Y296,301A,F298A}$ yeast cells (Fig. 3b).

Mutants of ERMES components lead to an altered phospholipid composition of mitochondria, in particular to decreased levels of the dimeric phospholipid cardiolipin[5,23,34,35]. The biosynthesis of cardiolipin occurs in the mitochondrial inner membrane, yet the precursor lipid phosphatidic acid and further phospholipids have to be transferred from the ER to mitochondria[11,42,43]. Currently, controversial views are discussed if ERMES plays a direct or indirect role in mitochondrial phospholipid homeostasis[5,6,8,17,44]. We used the *mdm10* site-specific mutants to determine whether the involvement of Mdm10 in phospholipid homeostasis was linked to ERMES or not. We labelled the yeast cells with [$^{33}$P]orthophosphate and analysed the phospholipid composition of mitochondria. The levels of cardiolipin were significantly diminished in Mdm10$^{Y296,301A,F298A}$ mitochondria, but not in the other mutant mitochondria (Fig. 3c).

Taken together, the binding of Mdm10 to ERMES is required for maintaining the tubular morphology and phospholipid homeostasis of mitochondria. Mdm10 mutants defective in binding to SAM or Tom7 show wild-type like mitochondrial

morphology and phospholipid levels, underscoring the specific functions of the Mdm10–ERMES connection.

**Mdm10 forms a mitochondrial membrane anchor of ERMES.** Mmm1, Mdm12 and Mdm34, but not Mdm10, belong to the tubular lipid-binding protein superfamily. They contain synaptotagmin-like mitochondrial lipid-binding protein (SMP) domains that can bind hydrophobic ligands and are thought to be involved in phospholipid transfer[45,46]. It has been assumed that Mdm34 is an integral protein of the mitochondrial outer membrane[11,14,42,46]. Since Mdm12 has been shown to bridge Mdm34 to the ER-integrated Mmm1 (ref. 46), the available results indicate that the Mdm34–Mdm12–Mmm1 assembly connects mitochondria to the ER and is involved in phospholipid transfer[5,8,34,35,45,46]. Thus, the current model does not leave any relevant role for Mdm10, raising the question why an impaired function of Mdm10 leads to ERMES-specific defects like described for the three SMP domain-containing proteins[5,13,19].

We directly compared the membrane localization of the four ERMES core components and observed an extraction of Mdm12

and Mdm34 at pH 11.5, whereas Mdm10 and Mmm1 remained in the membrane sheets (Fig. 4a, lanes 5 and 6). Thus, in contrast to the current assumption, Mdm34 does not behave as an integral membrane protein. Indeed, using five different prediction programmes (see Methods), we did not detect any transmembrane segment in Mdm34. In a previous study[14], Mdm34 was detected in the membrane fraction after carbonate extraction, however, milder extraction conditions (pH 11.0) were used compared with our analysis. Indeed, we found that at milder conditions, the ERMES subunits remained largely membrane-associated; only a fraction of Mdm34 was extracted (Fig. 4a, lanes 2 and 3). These results indicate that Mdm12 and Mdm34 behave as peripheral membrane proteins, which are associated with, but not fully integrated into the lipid phase. Mdm10 is the ERMES core component that is integrated into the lipid phase of the mitochondrial outer membrane.

It is unknown how Mdm10 associates with the three SMP domain-containing ERMES subunits. To determine the organization of ERMES, we systematically analysed the interaction of the ERMES core components on lysis of yeast cell extracts with non-ionic detergent. Tagged Mdm10 co-purified the SMP domain-containing ERMES subunits as expected (Fig. 4b, lane 7)[19]. On deletion of *MDM34*, the co-purification of Mdm12 and Mmm1 with Mdm10 was blocked, whereas deletion of *MDM12* or *MMM1* did not inhibit the interaction between Mdm10 and Mdm34 (Fig. 4b, lanes 8–10). These findings indicate a close association of Mdm10 and Mdm34. To obtain independent evidence, we used yeast strains with tagged Mmm1. Deletion of *MDM34* inhibited the co-purification of Mdm10 with tagged Mmm1, whereas deletion of *MDM10* attenuated but did not block the co-purification of Mdm34 with Mmm1 (Fig. 4c). Mixing experiments outlined in Supplementary Fig. 2d excluded the possibility that Mdm34 and Mdm10 may associate with tagged Mmm1 after cell lysis. A lack of Mdm12 blocked the interaction of Mdm10 and Mdm34 with Mmm1 (Fig. 4c). Taken together, we conclude that the ERMES core components form a chain of interactions consisting of Mdm10–Mdm34–Mdm12–Mmm1. Mdm10 supports the association of Mdm34 with Mdm12–Mmm1 but is not strictly required for this interaction.

Tom7 binds to Mdm10 released from the SAM complex[21,32,36], yet opposing views exist if Tom7 is part of the ERMES complex. Yamano *et al.*[21] reported that a fraction of Tom7 bound to Mdm10 may be partly associated with ERMES, however, subsequent studies did not detect Tom7 in the ERMES complex[15,16] and thus the current models of the ERMES complex do not include Tom7 as a stable subunit[11,42,47]. We systematically analysed if small Tom proteins are associated with ERMES by using yeast strains containing tagged Tom5, Tom6 or Tom7. On lysis with non-ionic detergent, only Tom7 pulled down ERMES subunits, whereas all three small Tom proteins pulled down the TOM complex (Fig. 4d, left panel), demonstrating that Tom7 is selectively associated with ERMES. Owing to its small size, Tom7 likely escaped detection in previous studies using mass spectrometry or gel electrophoresis. To determine how Tom7 interacts with ERMES, we used the Mdm10$^{G144L}$ and Mdm10$^{Y296A,F298A}$ site-specific mutants (Fig. 1a). In the Mdm10$^{G144L}$ mutant, Tom7 neither pulled down Mdm10 nor other ERMES subunits (Fig. 4d, right panel). In the Mdm10$^{Y296A,F298A}$ mutant, which is disturbed in the Mdm10–ERMES interaction, Tom7 associated with Mdm10, but not with other ERMES components. The site-specific *mdm10* mutants did not inhibit the association of Tom7 with the TOM complex (Fig. 4d, right panel), demonstrating that Tom7 was functional in the mutants. We conclude that the association of Tom7 with ERMES occurs via its binding to Mdm10.

Our findings lead to a new model of the organization of the ERMES complex. Mdm10 forms an integral membrane anchor of ERMES at mitochondria, whereas the peripheral membrane proteins Mdm34 and Mdm12 build the bridge to the ER-anchored Mmm1. In addition, Tom7 bound to Mdm10 is part of the ERMES complex.

**SAM-Mdm10 promotes outer membrane protein assembly**. Mdm10 is required for the efficient assembly of the TOM complex. Different models have been proposed to describe the role of Mdm10, either by promoting the assembly of the β-barrel protein Tom40 or by promoting the import of the receptor Tom22, which contains an α-helical transmembrane segment[18,21,31,32]. To determine which population of Mdm10 is involved in protein assembly, we studied TOM assembly in the *mdm10* site-specific mutants. The steady-state levels of the fully assembled TOM complex were decreased in Mdm10$^{Y73,75A}$ mitochondria, which are impaired in the Mdm10–SAM interaction, but not in Mdm10$^{G144L}$ and Mdm10$^{Y296A,F298A}$ mitochondria, which are defective in binding to Tom7 or ERMES, respectively (Fig. 5a). The assembly stages and kinetics of TOM complex formation can be directly monitored by studying the import of the $^{35}$S-labelled precursor of Tom40 into isolated mitochondria. The assembly of Tom40 occurs via two intermediates that can be resolved by blue native electrophoresis. Intermediate I represents binding of the precursor to the SAM complex in two stages, followed by formation of a Tom40 dimer (intermediate II) that assembles with Tom22 and small Tom proteins to form the mature TOM core complex[28,32,33]. TOM assembly was strongly diminished in Mdm10$^{Y73,75A}$ mitochondria, in particular formation of intermediate I and of the mature TOM complex was reduced, whereas in Mdm10$^{G144L}$, Mdm10$^{Y296A,F298A}$ and Mdm10$^{Y296,301A,F298A}$ mitochondria TOM assembly was not or only mildly affected (Fig. 5b,c). Thus, Mdm10 mutant mitochondria that are defective in the Mdm10–SAM interaction show a major defect in TOM assembly, but not Mdm10 mutants that are impaired in binding to Tom7 or ERMES.

The import pathway of the Tom40 precursor involves an initial translocation by the TOM complex to the intermembrane space side, followed by export via the SAM complex and assembly[27,28,41]. Since the levels of the TOM complex were reduced in Mdm10$^{Y73,75A}$ mitochondria, we asked if the inhibitory effect was caused by a defect in this initial translocation, that is, before the action of the SAM complex. However, the initial translocation of β-barrel precursors to a protease-protected location was only mildly reduced in Mdm10$^{Y73,75A}$ mitochondria (Fig. 5d), indicating that it was the subsequent SAM- and Mdm10-dependent steps that were mainly impaired in the mutant mitochondria. As control, the import of presequence-carrying preproteins via TOM and the presequence translocase of the inner membrane (TIM23) was not or only mildly affected in Mdm10$^{Y73,75A}$ mitochondria (Supplementary Fig. 3a). We conclude that the disturbed interaction of Mdm10$^{Y73,75A}$ with the SAM complex leads to an impaired assembly of the TOM complex.

It was reported that mitochondria lacking Mdm10 are impaired in the import of the Tom22 precursor into the mitochondrial outer membrane and that the SAM–Mdm10 complex, but not the SAM$_{core}$ complex, binds the precursor of Tom22 (refs 32,48). However, it has remained controversial if Mdm10 plays a role in Tom22 import or not (refs 18,31,32,48). We performed several assays to address a role of Mdm10 in Tom22 biogenesis. (i) Insertion of the Tom22 precursor into the outer membrane was analysed by treatment of mitochondria with

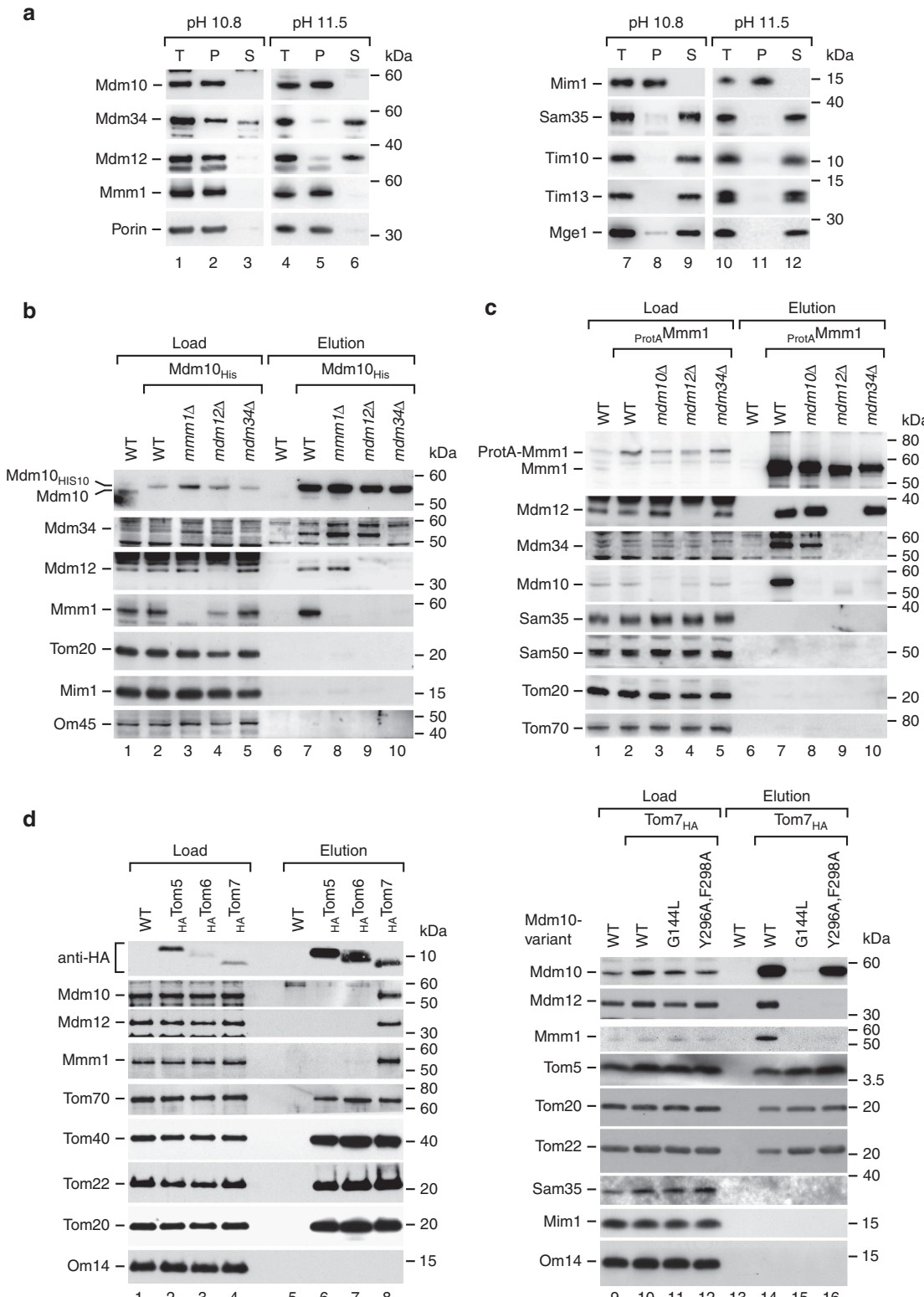

**Figure 4 | Mdm10 anchors the ERMES complex in the mitochondrial outer membrane.** (**a**) Mitochondria isolated from wild-type (WT) cells were incubated in 0.1 M Na$_2$CO$_3$ at pH 10.8 and 11.5. Soluble and membrane-bound proteins were separated by ultracentrifugation. Samples were analysed by SDS–polyacrylamide gel electrophoresis (SDS–PAGE) and immunodetection with the indicated antisera. Comparable amounts of the total sample (T), membrane pellet (P) and supernatant (S) were loaded. (**b**) Whole-cell extracts of WT, Mdm10$_{His}$, Mdm10$_{His}$ mdm12Δ, Mdm10$_{His}$ mdm34Δ and Mdm10$_{His}$ mmm1Δ cells were solubilized with digitonin and subjected to affinity purification using Ni-NTA. Samples were analysed by SDS–PAGE and immunodetection. Load 2%, elution 100%. (**c**) Whole-cell extracts of WT, $_{ProtA}$Mmm1, $_{ProtA}$Mmm1 mdm10Δ, $_{ProtA}$Mmm1 mdm12Δ and $_{ProtA}$Mmm1 mdm34Δ cells were solubilized with digitonin and subjected to affinity purification via IgG-Sepharose. Load 2%, elution 100%. (**d**) Mitochondria isolated from WT, $_{HA}$Tom5, $_{HA}$Tom6 and $_{HA}$Tom7 cells (left panel) or from WT and Tom7$_{HA}$ cells expressing the indicated mdm10 mutant forms (right panel) were solubilized with digitonin and subjected to affinity purification using anti-HA affinity matrix. Load 4% (left panel), 2% (right panel); elution 100%.

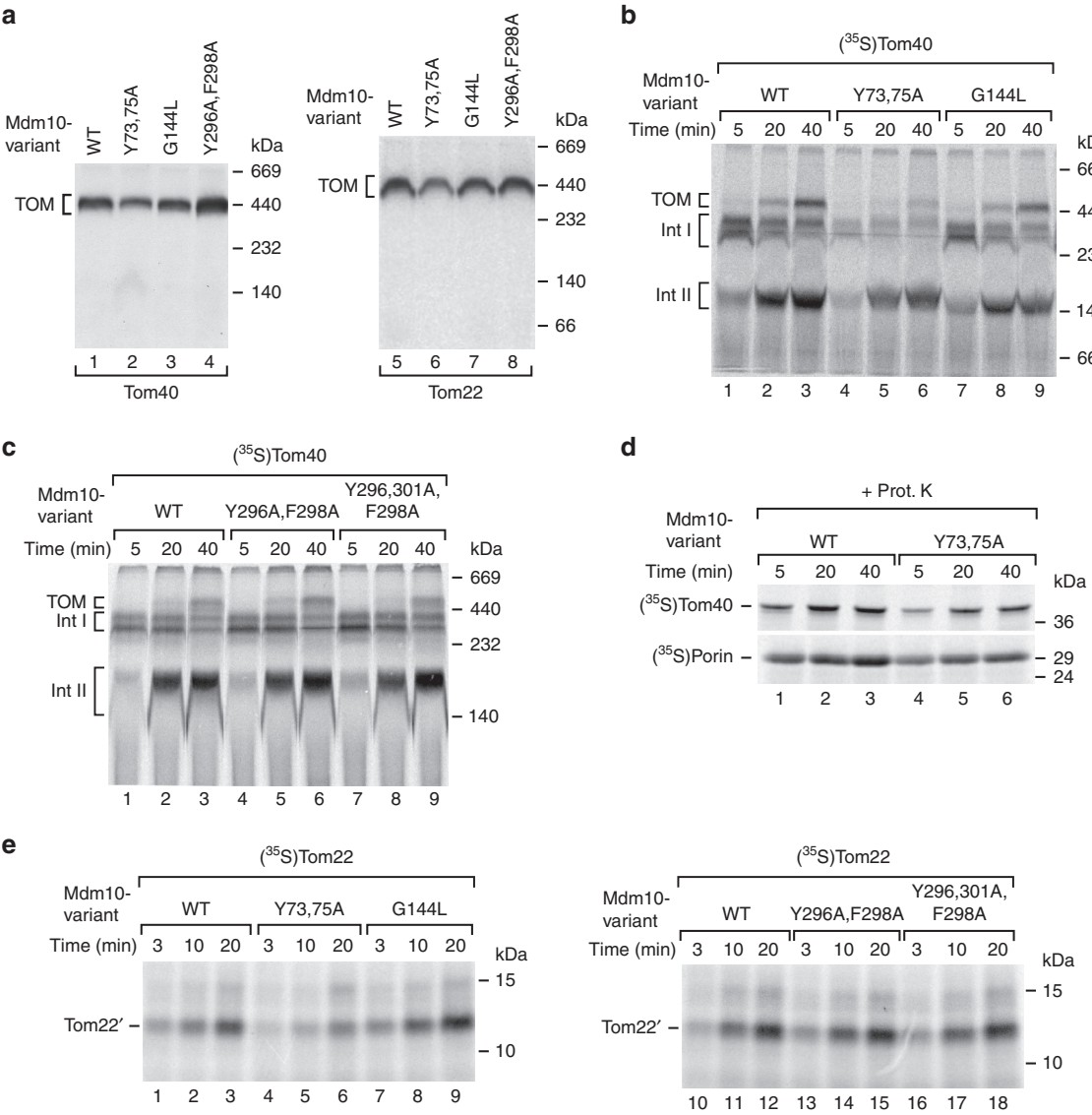

**Figure 5 | Mdm10 population bound to SAM mediates biogenesis of the TOM complex.** (**a**) Mitochondria isolated from cells expressing wild-type (WT) or mutant forms of *MDM10* were solubilized with digitonin. Protein complexes were separated by blue native electrophoresis. The TOM complex was detected by immunodecoration with antisera against Tom40 and Tom22. (**b**) [$^{35}$S]Tom40 was imported into mitochondria, which were isolated from cells expressing WT or mutant forms of *MDM10*, for the indicated periods. Mitochondria were reisolated, solubilized with digitonin and protein complexes were separated by blue native electrophoresis. Radiolabelled protein complexes were visualized by autoradiography. Int I/II, intermediate I/II. (**c**) [$^{35}$S]Tom40 was imported into isolated mitochondria as described in **b**. (**d**) [$^{35}$S]Tom40 and [$^{35}$S]Porin were imported into WT and Mdm10$^{Y73,75A}$ mitochondria for the indicated periods. The mitochondria were treated with proteinase K and analysed by SDS–polyacrylamide gel electrophoresis and autoradiography. (**e**) [$^{35}$S]Tom22 was imported into WT and the indicated Mdm10 mutant mitochondria, followed by treatment with proteinase K. The protease-protected fragment of membrane-inserted [$^{35}$S]Tom22 (Tom22′) is shown.

protease after the import reaction. The protease removes the N-terminal cytosolic receptor domain of Tom22 and generates a fragment, which consists of the single transmembrane fragment of Tom22 and its C-terminal intermembrane space domain[49]. Generation of the protease-protected fragment Tom22′ indicates a proper membrane insertion of the precursor. Membrane insertion of Tom22 was selectively impaired in Mdm10$^{Y73,75A}$ mutant mitochondria, but not in the other mutant mitochondria (Fig. 5e), indicating that SAM-bound Mdm10 is involved in the efficient import of Tom22 into the outer membrane. (ii) We asked if the Mdm10 β-barrel exhibits channel activity. We purified Mdm10 on recombinant expression and reconstituted it into liposomes. The liposomes were fused with a planar lipid bilayer for electrophysiological analysis. We observed a channel

activity with a main conductance $\bar{G}_{main} = 480\,pS$ and a slight cation selectivity of $P_K^+/P_{Cl}^- = 2.8{:}1$ (reversal potential $V_{rev} = 21.5\,mV$) with a functional unit of three independently gating pores (Fig. 6a,b). The main conductance is in a similar range as that of the Tom40 and Sam50 channels[27,50–52]. Addition of the full-length precursor of Tom22 led to a strong stimulation of the gating frequency of the channel (flickering) (Fig. 6c,d). Various control proteins, including the cytosolic domain of Tom22, Tom7 that binds to the outside of the Mdm10 β-barrel (Figs 1c and 4d)[21] and the multispanning outer membrane protein Om14 (ref. 53), did not affect the channel activity (Fig. 6e). The Tom22 precursor induced an increase of the maximal conductance of the Mdm10 channel to $\bar{G}_{main} = 550\,pS$ and a change of the functional unit to four independently gating

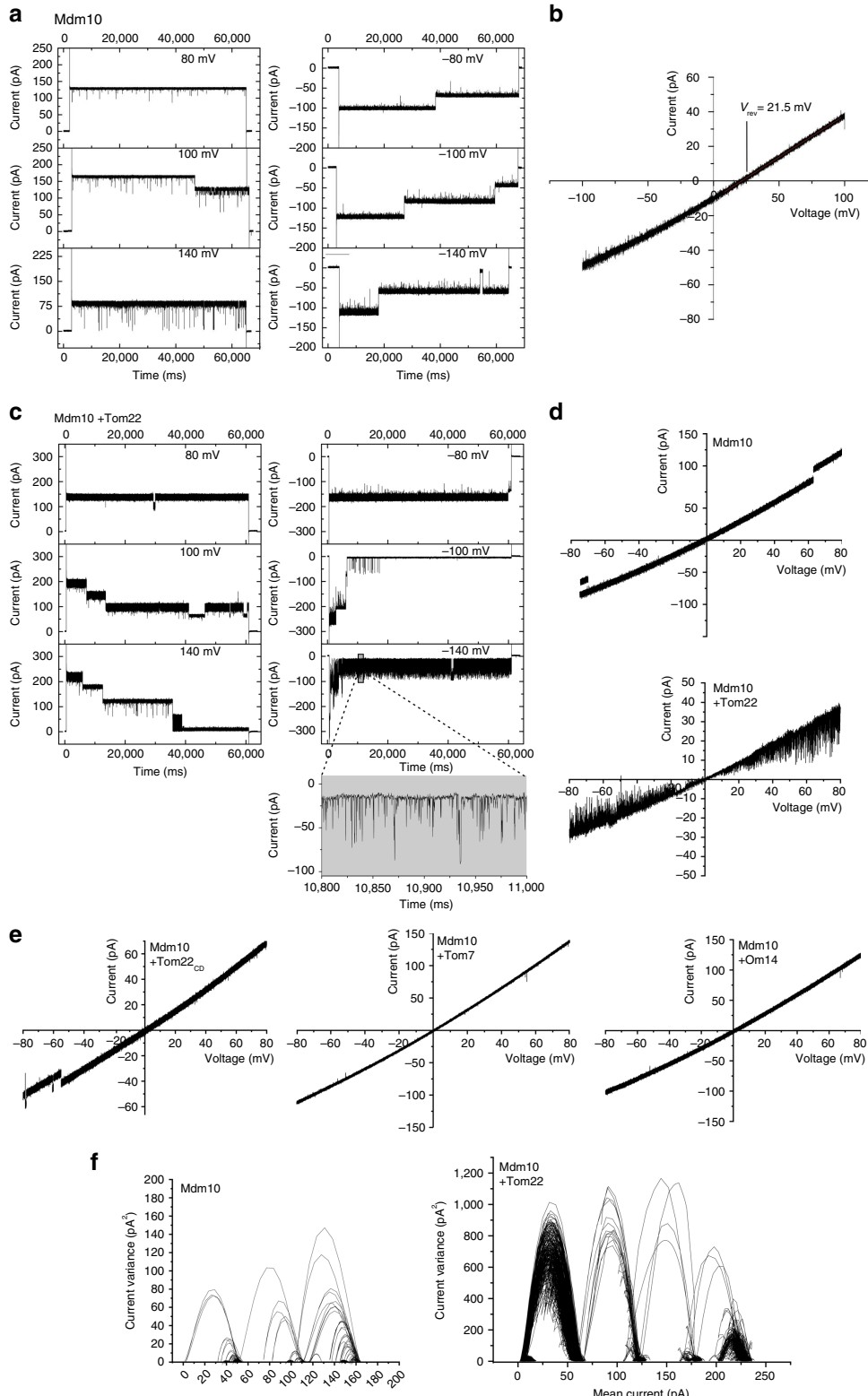

**Figure 6 | Mdm10 forms a Tom22-sensitive channel.** (**a**) Single-channel recording under symmetrical buffer conditions (250 mM KCl and 10 mM MOPS-Tris, pH 7.0) at the indicated $V_m$ from a planar lipid bilayer after a single fusion event of Mdm10 proteoliposomes. (**b**) Current–voltage relationship of Mdm10 (main conductance) in asymmetric buffer conditions (250/20 mM KCl (*cis/trans*) and 10 mM MOPS-Tris, pH 7.0). (**c**) Single-channel recording of Mdm10 under symmetrical buffer conditions as described in **a** after preincubation of Mdm10 proteoliposomes with full-length Tom22 precursor. (**d**) Current–voltage relationship of Mdm10 (main conductance) under symmetrical buffer conditions (upper panel). Mdm10 proteoliposomes were additionally preincubated with Tom22 (lower panel). (**e**) Current–voltage relationship of Mdm10 (main conductance) under symmetrical buffer conditions after preincubation of Mdm10 proteoliposomes with the cytosolic domain of Tom22 (Tom22$_{CD}$), Tom7 or Om14. (**f**) Mean-variance plot from a current recording at $V_m = 100$ mV of Mdm10 under symmetrical buffer conditions (left panel). Mdm10 proteoliposomes were additionally preincubated with Tom22 (right panel).

pores (Fig. 6f). (iii) Tom22 lacking the cytosolic domain was expressed as a fusion protein with glutathione *S*-transferase (GST) and purified. On incubation with lysed mitochondria, Tom22$_{\Delta N}$ pulled down Mdm10 (Supplementary Fig. 3b). Taken together, these results support a role of Mdm10 in the biogenesis of Tom22. Mdm10 forms a channel and the precursor of Tom22 stimulates the channel activity by increasing the gating frequency and channel conductance. Together with the binding of the Tom22 precursor to the SAM–Mdm10 complex *in organello*[32,48] and the analysis of Tom22 import and Tom40 assembly (Fig. 5), we conclude that SAM-bound Mdm10 promotes the biogenesis of both Tom22 and Tom40.

## Discussion

The mitochondrial outer membrane plays crucial functions in metabolite transport, protein biogenesis, lipid homeostasis and membrane dynamics[1,4,11,42,47,54,55]. The latter three functions have been linked to Mdm10. Since Mdm10 is present in both ERMES and SAM complexes[5,18,19,21,26], we aimed to dissect the functions of the two Mdm10 populations. We designed site-specific mutants that selectively inhibited the coupling of Mdm10 to either one of the complexes. We found that ERMES and SAM bind to opposite surfaces of the Mdm10 β-barrel and thus could separate protein biogenesis from lipid homeostasis and membrane morphology (Figs 1a and 7). The Mdm10–ERMES interaction is crucial for maintaining mitochondrial morphology and phospholipid homeostasis, whereas the Mdm10–SAM interaction is required for the biogenesis of outer membrane proteins.

The molecular function of Mdm10 has been unknown. We observed that it plays different roles in the ERMES complex and the SAM complex. In the ERMES core complex, Mdm10 forms an integral membrane anchor in the mitochondrial outer membrane. In contrast to previous assumptions[14], Mdm34 does not contain a transmembrane anchor in the outer membrane, but behaves as a peripheral membrane protein like Mdm12, whereas the β-barrel of Mdm10 is integrated into the lipid phase of the outer membrane. Suresh *et al.*[56] observed that on prolonged glucose starvation the localization of a large fraction of GFP-tagged Mdm34 shifted from ERMES foci to the cytosol in a reversible manner, supporting the view that Mdm34 is only peripherally attached to the mitochondrial outer membrane.

It has been unclear why Mdm10, which does not contain a lipid-binding domain, is required for mitochondrial phospholipid homeostasis like Mdm34, Mdm12 and Mmm1, which all contain lipid-binding SMP domains[5,45,46]. The architecture of the ERMES complex suggests that the membrane anchor function of Mdm10 is crucial for the function of ERMES, however, it is open if in addition Mdm10 and hydrophobic patches inside its β-barrel channel[38] may play a direct role in lipid homeostasis. Tan *et al.*[23] showed that overexpression of the mitochondrial outer membrane protein Mcp1 restored the mitochondrial morphology of cells lacking Mdm10, however, cardiolipin levels, protein import into mitochondria and cell growth at higher temperature were not fully rescued, indicating that crucial functions of Mdm10 were not suppressed by Mcp1; Mcp1 is not associated with ERMES[23] and its molecular function is unknown. Interestingly, Kornmann *et al.*[5] showed that an artificial tethering construct, which connects ER and mitochondria, restores the mitochondrial morphology of cells lacking Mdm12 or Mdm34, but not of cells lacking Mmm1 or Mdm10. Thus, the artificial tether can replace functions of the two peripheral membrane proteins Mdm12 and Mdm34, but not that of the integral membrane proteins Mmm1 and Mdm10 in line with our view that Mdm12 and Mdm34 form the bridge between Mmm1 and Mdm10. Mmm1 and Mdm10 likely perform additional functions that cannot be substituted for by an artificial tether. Taken together, we favour the model that the ERMES complex consists of a chain of interactions from membrane integrated Mdm10 in the mitochondrial outer membrane via the peripheral subunits Mdm34 and Mdm12 to Mmm1 in the ER (Fig. 7).

In the SAM complex, Mdm10 is required for promoting the assembly of the TOM complex. Our analysis of Mdm10 concludes the controversial discussion if it promotes the biogenesis of either Tom40 or Tom22 (refs 18,31,32) by demonstrating that Mdm10 is required for the biogenesis of both precursor proteins. First, the site-specific *mdm10* mutant with a defect in binding to SAM impairs assembly of Tom40. The area of the Mdm10 β-barrel surface that we identified as binding site for SAM is highly conserved on the Tom40 β-barrel (Supplementary Fig. 4)[38]. Yamano *et al.*[31] proposed that Mdm10 promotes the release of the Tom40 precursor from SAM and thus supports its assembly into the TOM complex. Since the complete β-barrel is folded in association with SAM[41], the conserved SAM-binding site of Tom40 will be formed.

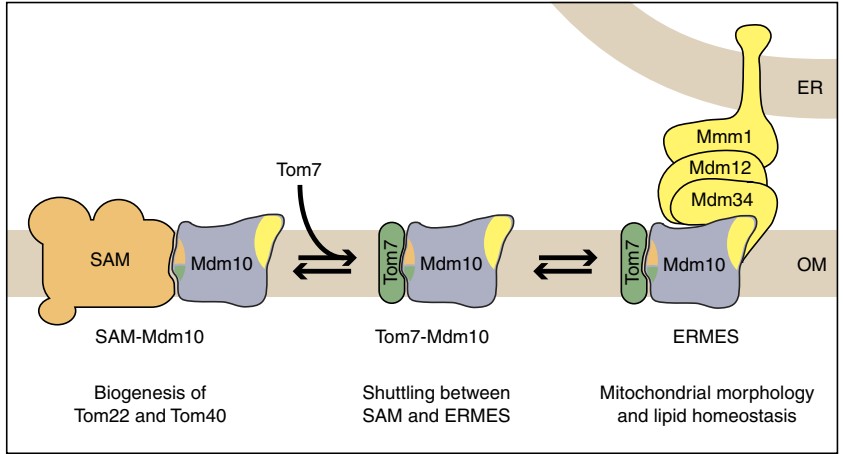

**Figure 7 | Hypothetical model of the dynamic interaction of Mdm10 with partner proteins.** Mdm10 interacts with SAM and ERMES via opposite surfaces of its β-barrel domain. Mdm10 forms an integral mitochondrial membrane anchor of ERMES and is required for ERMES functions in maintaining mitochondrial morphology and lipid homeostasis. SAM-bound Mdm10 promotes biogenesis of the TOM complex. The Mdm10-binding sites for Tom7 and SAM are in close proximity. Thus, Tom7 binds only to Mdm10 released from SAM and remains bound to Mdm10 in the ERMES complex. Not shown is the accessory subunit Gem1 that associates with ERMES in substoichiometric amounts. OM, outer mitochondrial membrane.

Binding of Mdm10 to SAM prevents a re-binding of folded Tom40 to SAM and thus favours an efficient displacement of Tom40 from SAM. Second, the precursor of Tom22 binds to the Mdm10-bound SAM complex *in organello*, not to the Mdm10-free SAM$_{core}$ complex[32]. However, it has been open if Mdm10 plays a direct or indirect role in the biogenesis of Tom22 (refs 18,31). We show that purified Mdm10 forms a β-barrel channel, which is specifically stimulated by addition of the precursor of Tom22. Insertion of Tom22 with its α-helical transmembrane segment into the outer membrane is impaired in the Mdm10 mutant defective in binding to SAM. Taken together, we conclude that SAM-bound Mdm10 functions in the biogenesis of both Tom22 and Tom40.

Our study also reveals why Tom7 can only bind to SAM-free Mdm10 (refs 21,32). Binding to Tom7 involves a conserved glycine residue of Mdm10 that is located in immediate vicinity to the SAM-binding site (Fig. 1a), suggesting a spatial overlap of the binding regions. Tom7 thus binds to Mdm10, which has been released from the SAM complex, and favours the shuttling of Mdm10 to ERMES[21,32]. Since ERMES binds to the opposite surface of the β-barrel, Tom7 remains bound to Mdm10 at ERMES and becomes the sixth subunit of ERMES. When the levels of Tom7 are increased, the interaction of Mdm10 with SAM is diminished and its association with ERMES is moderately increased[21,32]. On deletion of *TOM7*, Mdm10 is redistributed from ERMES to SAM, resulting in moderately reduced levels of Mdm10–ERMES[21,32]. The mitochondrial phospholipid profile is unaltered, but the mitochondrial morphology is disturbed in the absence of Tom7 (refs 21,36). Since the mitochondrial morphology is affected in other *tom* mutant strains as well[18,57], this phenotype may be linked to impaired TOM function. Indeed, our site-specific *mdm10* mutant specifically affects the binding of Tom7 to Mdm10, whereas Tom7 is still present in the TOM complex to warrant proper protein import and the mitochondrial morphology is not affected. We conclude that unlike Mdm10, Tom7 is not crucial for ERMES function, and a destabilization of Tom7 association with Mdm10 and ERMES in the site-specific mutants does not lead to major defects. Tom7 thus behaves as a non-essential regulatory subunit of ERMES like Gem1 (refs 15–17).

In summary, we can assign four distinct molecular functions to the β-barrel of Mdm10: (i) binding to Mdm34 and functioning as integral membrane anchor of ERMES; (ii) binding to the SAM complex via a region that is also conserved in Tom40, explaining how Mdm10 favours the release of folded Tom40 from SAM; (iii) formation of a channel that is sensitive to the precursor of Tom22; and (iv) binding to Tom7, which interacts with Mdm10 released from SAM. Importantly, since ERMES and SAM bind to opposite surfaces of Mdm10, we can separate ERMES- and SAM-specific functions of Mdm10 and thus can assign the maintenance of mitochondrial morphology and lipid homeostasis to ERMES, whereas protein assembly is linked to Mdm10 bound to the SAM complex.

## Methods

**Yeast strains.** The *Saccharomyces cerevisiae* strains used in this study are listed in Supplementary Table 1. The point mutations were introduced in the *MDM10* open reading frame (ORF) inserted into a pFL39 plasmid by site-directed mutagenesis. Plasmid shuffling was used to generate yeast strains expressing the *mdm10* mutant forms or the corresponding wild-type *MDM10* (refs 38,41). In the shuffle strain the chromosomal copy of *MDM10* was deleted with an ADE2 marker and the pFL39 plasmids encoding *mdm10* mutants were transformed in the presence of a YEp352 plasmid encoding wild-type *MDM10* and a URA3 selection marker. Yeast cells that have lost YEp352 encoding *MDM10* were selected by growth on 5-fluoroorotic acid-containing medium.

To generate yeast strains expressing His-tagged Mdm10, genetic information encoding a deca His-tag was introduced before the stop codon of *MDM10* in the pFL39 plasmid. The corresponding yeast strains were generated by plasmid

shuffling. To obtain Mdm10 mutant strains expressing $_{ProtA}$Mmm1 or Tom7$_{HA}$, we chromosomally introduced the genetic information coding for the affinity tag in the *MDM10* shuffle strain. Subsequently, plasmid shuffling generated yeast strains expressing the Mdm10 mutant forms. Sam50 was tagged with a protein A tag directly in the Mdm10 mutant strains. The genetic information for a triple-haemagglutinin tag was chromosomally integrated in front of the stop codon of *TOM7*. For yeast strains expressing *SAM50* or *MMM1* fused to an N-terminal protein A tag, a cassette coding for *HIS3MX6-NOP1-ProtA-TEV* was chromosomally integrated in front of the *SAM50* ORF or *MMM1* ORF[16,38,41]. Genetic information encoding a deca-His tag was chromosomally integrated in front of the stop codon of the *MDM10* ORF in the *mdm12Δ*, *mdm34Δ* or *mmm1Δ* strains by homologous recombination. Similarly, the nucleotide sequence of *HIS3MX6-NOP1-ProtA-TEV* was chromosomally integrated in front of the *MMM1* ORF in the *mdm10Δ*, *mdm12Δ* or *mdm34Δ* strains by homologous recombination. For the N-terminal tagging of small Tom proteins, the ORFs encoding *TOM5*, *TOM6* or *TOM7* were deleted with an URA3 marker. Subsequently, the strains were transformed with a PCR product coding for *TOM5*, *TOM6* or *TOM7* and a triple HA-tag after the start codon. *HA-TOM5*, *HA-TOM6* and *HA-TOM7* were inserted into their native locus by homologous recombination and replaced the URA3 marker. Transformants were selected by growth on 5-fluoroorotic acid-containing medium.

**Growth conditions and isolation of mitochondria.** For biochemical analysis, *mdm10* mutant strains, strains expressing HA-tagged small Tom proteins and their corresponding wild-type strains were grown on YPG medium (glycerol as carbon source)[19] at 30 °C to an early exponential growth phase. For biochemical and fluorescence microscopy experiments involving the Mdm10$^{Y296,301A,F298A}$ mutant, all mutant and wild-type cells were first grown in YPG medium at 30 °C to mid-log phase and then shifted to 37 °C for 3 h. Deletion strains of ERMES mutants were grown on YPS (sucrose as carbon source) at 24 °C. Mitochondria were isolated from yeast cells by differential centrifugation. Yeast cells were incubated with 10 mM dithiothreitol in 100 mM Tris/H$_2$SO$_4$, pH 9.4, for 30 min at 30 °C. After washing with 1.2 M sorbitol, 20 mM KP$_i$, pH 7.4, yeast cells were treated with 3 mg zymolyase per g cells for 45 min at 30 °C to generate spheroplasts. Spheroplasts were washed with 1.2 M sorbitol, 20 mM KP$_i$, pH 7.4, and resuspended in homogenization buffer (0.6 M sorbitol, 10 mM Tris/HCl, pH 7.4, 1 mM EDTA, 0.2% [w/v] bovine serum albumin and 1 mM phenylmethyl sulfonyl fluoride). The cell membrane was ruptured by homogenization of the spheroplasts with 15 strokes using a glass-Teflon homogenizer. Subsequently, cell debris and nuclei were removed by centrifugation (2,000$g$, 5 min, 4 °C). The supernatant was subjected to a second centrifugation step (17,500$g$, 15 min, 4 °C) to pellet mitochondria. The mitochondrial pellet was washed in SEM buffer (250 mM sucrose, 1 mM EDTA and 10 mM MOPS-KOH, pH 7.2). Finally, mitochondria were resuspended in SEM buffer at a protein concentration of 10 mg ml$^{-1}$ and shock-frozen in liquid nitrogen. Samples were stored at − 80 °C until use. The mutant mitochondria analysed contained an intact outer membrane like wild-type mitochondria.

**Preparation of total cell extract.** For the preparation of total cell extracts, yeast cells were grown to an early logarithmic growth phase and washed twice with water and with lysis buffer (20 mM Tris-HCl, pH 7.4, 0.1 mM EDTA, 50 mM NaCl and 10% glycerol). Subsequently, cells were shock-frozen in liquid nitrogen and grinded using a cryomill (Retsch) at 25 Hz for 10 min. Cell powder was stored at − 80 °C until use.

**Protein import into isolated mitochondria.** The precursor proteins for *in vitro* import were synthesized in a coupled transcription/translation system based on reticulocyte lysate (TNT SP6 quick-coupled transcription/translation kit; Promega). pGEM4z plasmids or RNA encoding proteins of interest were added to the reaction. The proteins were labelled with [$^{35}$S]methionine. For *in vitro* import reactions, ∼ 5–10% (v/v) of translation lysate were used. The standard import reaction[41] was performed in import buffer (3% (w/v) bovine serum albumin, 250 mM sucrose, 80 mM KCl, 5 mM MgCl$_2$, 5 mM methionine, 2 mM KH$_2$PO$_4$ and 10 mM MOPS-KOH, pH 7.2) supplemented with 4 mM ATP and 4 mM NADH. Import reactions were stopped by addition of a mixture of 8 μM antimycin A, 1 μM valinomycin and 20 μM oligomycin (final concentrations). In case of outer membrane precursor proteins, transfer on ice stopped the import reaction. To analyse the membrane insertion of [$^{35}$S]Tom22 we imported a Tom22 variant, which contained three additional methionines at the C terminus. After the import reaction, mitochondria were incubated with proteinase K (50 μg ml$^{-1}$ final concentration) for 15 min on ice. The protease was inhibited by addition of phenylmethyl sulfonyl fluoride.

The samples were analysed by SDS–PAGE or blue native electrophoresis, followed by autoradiography. For native analysis, the samples were solubilized with lysis buffer (20 mM Tris-HCl, pH 7.4, 0.1 mM EDTA, 50 mM NaCl and 10% glycerol) containing 1% (w/v) digitonin for 15 min on ice. Non-solubilized material was removed by centrifugation and the supernatant was loaded on a blue native gel.

**Affinity purification from total cell extracts.** For affinity purification out of total cell extracts, yeast strains expressing protein A-tagged Mmm1 or Sam50, and yeast

strains expressing His-tagged Mdm10 were used. The cell extract was prepared by cryo-grinding. For affinity purification via the protein A tag, the cell powder was solubilized with lysis buffer containing 1% (w/v) digitonin (1 ml lysis buffer per 100 mg cells) for 45 min at 4 °C under constant rotation. After removal of non-solubilized material, the sample was incubated with IgG-Sepharose (GE Healthcare) for 90 min. Unbound material was removed by centrifugation. The beads were washed with an excess of lysis buffer containing 0.1% (w/v) digitonin. Bound proteins were eluted by incubation with His-tagged AcTEV Protease (Invitrogen) in lysis buffer containing 0.1% (w/v) digitonin for SDS–PAGE analysis or 1% (w/v) digitonin for blue native electrophoresis. The elution step was performed overnight at 4 °C under constant shaking. The His-tagged AcTEV Protease was removed by adding Ni-NTA and further incubation for 30 min at 4 °C.

For affinity purification of His-tagged Mdm10, the cell powder was solubilized with lysis buffer containing 1% (w/v) digitonin and 10 mM imidazole (1 ml lysis buffer per 100 mg cells) for 45 min at 4 °C under constant rotation. After removal of insoluble material, the cell extract was incubated with Ni-NTA for 90 min and purification was continued as described for lysed mitochondrial extracts.

**Affinity purification from lysed mitochondrial extracts.** For purification of His-tagged Mdm10-containing complexes, mitochondria were lysed with lysis buffer containing 1% (w/v) digitonin and 10 mM imidazole for 15 min on ice. Insoluble material was removed by centrifugation and the remaining lysate was incubated with Ni-NTA agarose for 60 min at 4 °C under constant shaking. Unbound samples were removed and the beads were washed with an excess of lysis buffer containing 0.1% (w/v) digitonin and 20 mM imidazole. Bound proteins were eluted with lysis buffer containing 250 mM imidazole and 0.1% (w/v) digitonin for SDS–PAGE studies or 1% (w/v) digitonin for blue native electrophoresis.

Protein complexes containing HA-tagged small Tom proteins were purified from isolated mitochondria utilizing an HA-affinity tag. Mitochondrial membranes were lysed by incubation with lysis buffer containing 1% (w/v) digitonin for 15 min at 4 °C. Insoluble material was removed by centrifugation. The supernatant was incubated with anti-HA affinity matrix (Roche) for 1 h at 4 °C under constant shaking. Unbound material was removed by centrifugation and the beads were washed extensively with lysis buffer containing 0.1% (w/v) digitonin. Bound proteins were eluted by incubation with SDS sample buffer.

For co-immunoprecipitation, antibodies against Mdm10 were coupled to protein A-sepharose with 7 mM dimethylpimelidate in 0.1 M sodium tetraborate for 30 min at room temperature. Mitochondria were lysed by incubation with lysis buffer containing 1% (w/v) digitonin for 15 min at 4 °C. Insoluble material was removed by centrifugation. The supernatant was incubated with anti-Mdm10 coupled Protein A-Sepharose CL-4b (GE Healthcare) beads for 1 h at 4 °C under constant shaking. Unbound material was removed by centrifugation, followed by extensive washing of the beads with lysis buffer containing 0.1% (w/v) digitonin. Bound proteins were eluted by 0.1 M glycine/HCl, pH 2.5. Samples were immediately neutralized with TRIS base and precipitated for SDS–PAGE analysis.

GST fusion constructs consisting of GST, a linker containing a Thrombin cleavage site, and Tom22$_{\Delta N}$ (amino-acid residues 85–152) or Tom5$_{\Delta N}$ (amino-acid residues 16–50) were expressed in *Escherichia coli* and coupled to Glutathione Sepharose 4b (GE Healthcare) beads[52]. Mitochondria were lysed by incubation in GST buffer (20 mM HEPES-KOH, pH 7.5, 10 mM Mg-acetate, 100 mM K-acetate and 10% (v/v) glycerol) containing 1% (w/v) digitonin for 15 min at 4 °C. Insoluble material was removed by centrifugation. The supernatant was incubated with Glutathione Sepharose 4b beads coated with GST-Tom22$_{\Delta N}$, GST-Tom5$_{\Delta N}$ or GST at 4 °C under constant shaking. Unbound material was removed, followed by extensive washing with GST buffer containing 0.5% (w/v) digitonin. The GST fusion proteins were cleaved overnight at 4 °C by incubation in GST buffer supplemented with 0.5% (w/v) digitonin, 2.5 mM CaCl$_2$ and 50–80 U ml$^{-1}$ thrombin under constant shaking (elution).

**Carbonate extraction.** Mitochondria were incubated with 0.1 M Na$_2$CO$_3$, pH 10.8–11.5, for 30 min at 4 °C. Membranes were separated from the supernatant by ultracentrifugation (136,000*g*, 30 min, 4 °C). Total samples, membrane pellet and supernatant fractions were precipitated with trichloroacetic acid and analysed by SDS–PAGE.

**Microscopy.** The mitochondrial morphology was studied by fluorescence microcopy based on the staining of mitochondria with the fluorescent dye DiOC$_6$ (Invitrogen). Yeast cells were pre-cultivated to mid-exponential phase in YPG medium at 30 °C, and prior imaging were shifted to 37 °C for 3 h. To stain mitochondria, yeast cells were incubated with DiOC6 following the manufacturer's instructions. DiOC$_6$ is taken up by the mitochondria in a membrane potential-dependent manner. We used the Olympus BX61 microscope equipped with the UPLFLN × 100/1.3 objective (Olympus) and F-view charge-coupled device camera (Soft Imaging System) to study the fluorescently labelled cells[58]. The fluorescence of DiOC6 was visualized using a 470/40 nm bandpass excitation filter, a 495 nm dichromatic mirror and a 525/50 nm bandpass emission filter. Z-stack images were collected with 0.5 μm intervals along the Z axis and analysed with the Cell-P software (Olympus). For quantification, 300 cells of three independent cultures (at least 50 cells per culture) of each strain were analysed.

**Phospholipid analysis.** To determine the phospholipid content, yeast cells were grown on YPG and shifted to 37 °C for 3 h. Subsequently, cells corresponding to 25 OD$_{600}$ were centrifuged and resuspended in a small volume of YPG (5 OD$_{600}$ per ml). Yeast cells were labelled with 190 μCi [$^{33}$P]orthophosphate for 90 min at 37 °C. Yeast cells were pelleted, washed with water and mitochondria were isolated by differential centrifugation. Phospholipids were extracted with a 2:1 (v/v) mixture of chloroform/methanol. One-dimensional thin-layer chromatography was performed with chloroform/ethanol/water/triethylamine (30/35/7/35, v/v) as mobile phase[59]. Radiolabelled phospholipids were visualized by autoradiography. Phospholipid species were identified by co-migration of phospholipid standards (Avanti). Single phospholipids were quantified with ImageQuant (GE Healthcare).

**Electrophysiological measurements.** Mdm10$_{His}$ was recombinantly expressed in *E. coli* BL21 (DE3) RIL cells. *E. coli* cells were grown at 37 °C in LB medium to an OD$_{600}$ of 0.6. Subsequently, isopropyl-β-D-thiogalactoside (1 mM final concentration) was added to induce expression of Mdm10 for 4 h at 37 °C. Cells were collected, lysed under denaturing conditions and soluble cell debris was removed by centrifugation. Mdm10$_{His}$ was purified via Ni-NTA agarose and eluted in elution buffer (6 M urea, 100 mM NaH$_2$PO$_4$ and 10 mM Tris/HCl, pH 5.9). For reconstitution, liposomes containing a lipid mixture (50% L-α-phosphatidylcholin, 33% L-α-phosphatidylethanolamin, 10% L-α-phosphatidylinositol, 2% L-α-phosphatidylserin and 5% cardiolipin) were incubated with urea-denatured Mdm10 in the presence of 1% (w/v) SDS. Detergent and urea were removed by dialysis and incubation with Calbiosorb adsorbent[60]. The channel activity was measured in a planar lipid bilayer and analysed as described[60]. $\bar{G}_{main}$ values of the Mdm10 channel were calculated from the linear slope of current voltage curves between ± 30 mV from bilayer containing a single active Mdm10 channel unit. We used mean-variance plots to calculate the number of gating pores. Mean-variance plots were generated as described[60,61] using at least five current recordings from three independent preparations.

Tom7$_{His}$, Tom22$_{His}$ and Om14$_{His}$ were synthesized in a wheat germ-based cell-free translation system (5Prime)[32]. Proteins were purified under denaturing conditions. The cytosolic domain of Tom22 fused to a His-tag was produced in *E. coli* and purified as described[41]. The proteins were incubated with Mdm10-containing proteoliposomes for 30 min on ice, followed by electrophysiological analysis[60].

**Miscellaneous.** Five prediction programmes were used to search for putative transmembrane segments of Mdm34: DAS transmembrane prediction server (http://www.sbc.su.se/∼miklos/DAS/; 1997); HMMTOP Prediction of transmembrane helices and topology of proteins Version 2.0 (http://www.enzim.hu/hmmtop/; 2001); TMHMM Server v. 2.0 Prediction of transmembrane helices in proteins (http://www.cbs.dtu.dk/services/TMHMM-2.0/; 1998); TMpred Prediction of transmembrane regions and orientation (http://www.ch.embnet.org/software/TMPRED_form.html; 1993); and SOSUI Classification and secondary structure prediction of membrane proteins (http://harrier.nagahama-i-bio.ac.jp/sosui/; 1998). The homology models of *Saccharomyces cerevisiae* Mdm10 and Tom40 were derived from Flinner *et al.*[38].

Proteins were transferred from blue native gels and SDS-containing gels to polyvinylidene difluoride membranes (EMD Millipore) by semi-dry western blotting. Proteins were detected with specific antibodies (Supplementary Table 2), which were tested against mitochondria from the corresponding mutant strains. Enhanced chemiluminescence was used to detect the immunospecific signals[62]. We used X-ray films (Medix XBU, Foma) or the image reader LAS3000 (FujiFilm) to detect the immunosignals. The STORM phosphoimager system was used to detect $^{35}$S-labelled proteins by autoradiography. Non-relevant lanes were digitally removed, indicated by separating lines. Uncropped versions of all important immunoblots and gels are presented in Supplementary Figs 5 and 6.

**Data availability.** The authors declare that data supporting the findings of this study are available within the article and its Supplementary Information. All the other data can be obtained from the corresponding authors upon reasonable request.

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

## Acknowledgements

We thank N. Zufall for expert technical assistance. Work included in this study has also been performed in partial fulfilment of the requirements for the doctoral theses of L.E., L.B., V.K., S.P.S. and the bachelor thesis of K.E. This work was supported by the Sonderforschungsbereiche 746 and 1140, the Deutsche Forschungsgemeinschaft (BE 4679/2-1, ME 1921/4-1, PF 202/8-1, WA 681/2-1), the Excellence Initiative of the German federal and state governments (EXC 294 BIOSS; GSC-4 Spemann Graduate

School), the European Research Council (ERC) Consolidator Grant No. 648235 and an EMBO long-term fellowship (to Ł.O.).

## Author contributions

L.E., Ł.O., L.B., V.K., O.M., S.P.S., K.E., N.F., S.B.S., B.G. and T.B. performed the experiments and analysed data together with C.M., N.W., E.S., R.W. and N.P.; T.B., N.P., E.S., R.W. and L.E. designed and supervised the project; L.E., Ł.O., O.M., S.P.S., V.K., R.W. and T.B. prepared the figures; N.P. and T.B. wrote the manuscript; all authors discussed results from the experiments and commented on the manuscript.

## Additional information

**Competing financial interests:** The authors declare no competing financial interests.

