## [Peer review file · Nature Communications]

Reviewer #1 (Remarks to the Author)

This is a nice manuscript on a very complex question: how an Eucariotic cell is structurally built up and more specifically how mitochondria is connected with the ER. Here the authors add additional evidence which of a set of four putative protein is responsible for the interaction. Caused by the lack of high resolution structure the interaction is basically shown by biochemical assays. Previous publications revealed a set of proteins essential for the function, here the authors could show binding areas and subsequently suggest the strongest interacting proteins.

In absence of structural information a combination of methods allows to reveal interaction, In my opinion while waiting for the structure this is the best what one can do.

Knowledge on this interaction is fundamental for the understanding of cellular function.

The experiment are carefully done and described sufficiently.

The manuscript is partly difficult to read and to get the novelty. In the result section previous results are not clearly separated from novel once. The novelty would be more aparent if a number of comments are places into the introduction. Also, in this section I do not expect to review all previous works but rather one reference to an earlier work on which the result is based. Somewhat difficult for outsider to read a number of previous publication to understand the validity of an evaluation of a finding.

Reviewer #2 (Remarks to the Author)

A beta barrel protein, Mdm10, is located in the outer membrane and is associated with the ERMES complex, which connects the ER and mitochondria, and the SAM complex, which imports proteins into the mitochondria, in yeast. Based on this dual association of Mdm10 with the two functionally distinct complexes, it has been suggested that ERMES and SAM are functionally coordinated.

Ellenrieder et al. substituted conserved residues in Mdm10 and created mutants that were defective specifically in interactions with ERMES or SAM. Using these novel mutants, the authors conclusively demonstrate that ERMES-associated Mdm10 anchors ERMES to the mitochondrial outer membrane; SAM-associated Mdm10 is required for the assembly of the TOM complex.

Therefore, the roles of Mdm10 in ERMES and SAM are separable and likely independent.

Interestingly, three-dimensional structural modeling predicts that the two groups of mutations are located on the opposite sides of the beta barrel; Mdm10 appears to possess two functional interfaces. SAM-Mdm10 has been shown to be important for importing Tom22 into mitochondria. Ellenrieder et al. show that purified Mdm10 exhibits a Tom22-stimulated channel activity in the lipid bilayer, suggesting that Mdm10 forms a protein channel for Tom22 import. These exciting findings clearly separate two functions of Mdm10 in protein import and organelle interaction. The experiments are extensively performed, and the necessary controls are included. I accordingly recommend that this manuscript be published in Nature Communications.

Minor comments

1. The authors propose that Mdm10 shuttles between ERMES and SAM. However, the data are also consistent with another model in which two different populations of Mdm10 stably associate with ERMES or SAM without exchange. Tom7 may help prevent Mdm10 from being recruited exclusively to SAM during the assembly of ERMES.

2. The authors reconstituted purified Mdm10 into liposomes. Can the proteoliposomes import Tom22?

Reviewer #3 (Remarks to the Author)

This is an important work that provides information on the long-discussed role of Mdm10. The lack of Mdm10 leads to multiple defects related to mitochondrial morphology, lipid biogenesis, and biogenesis of outer membrane proteins. These various functions are executed through differential interactions with Mdm10 partners, such as the ERMES and SAM complexes and the Tom7 protein.

Based on the structural beta-barrel prediction and subsequent genetics the authors aimed to dissect the relevance of various interactions. This approach turned out to be a full success. The authors found the specific binding sites and could uncouple the Mdm10 individual functions caused by differential interactions.

This work provides significant progress in the understanding of processes that contribute to various aspects of mitochondrial biology. This, it is of interest to specialists from multiple fields of mitochondrial research as well as for general audience. The conclusions are well supported by the data of high quality and overall the entire story is extremely convincing.

A few points require experimental clarification and/or discussion

- In Fig. 1 and in many other places either the double mutant in the aromatic string (YF) or the triple mutant (YYF) is used. For example the growth test is performed on the triple mutant (Fig. 1b), and complex analyses for the double mutant (Fig. 1c), and this goes on in the subsequent experiments. Figure 2C may suggest that the double mutant has a milder phenotype in comparison with the more robust triple mutant. This point should be clarified with explicit conclusions. The differences in the interaction between Mmm1 and the Mdm10 mutants (Fig. 2C), especially the double mutant, should be quantified on the basis of multiple experiments and statistical significance of differences should be analyzed.
- It will be useful to clarify, either experimentally or by providing a reference, whether the lack of Tom7 affects the ERMES architecture and function.
- Figure 5 is a little bit confusing. I agree with the authors' conclusions however I feel that the data presentation can be improved. The moderate decrease in the TOM complex correlates well the import efficiency of the mitochondria from the Y73,75A mutant. Thus, the defect in the translocation more or less reflects a mild problem with the TOM translocase. However, the assembly defect of the Tom40 substrate is much bigger, justifying in full the authors' conclusion on the importance of this Mdm10 site for SAM contacts. I find the control imports of presequence precursors and the entire discussion on the stoichiometry of TOM and TOM complexes dispensable. Negative controls are in general very useful, but Fig. 5e can be easily moved to the Supplement here following shortening of the text on this aspect.
- The interesting electrophysiology experiments should be complemented by the analysis of Tom22 import into the mutants of Mdm10 that affect ERMES, SAM and Tom7 interactions.

Further points:

- I suggest considering a slight change in the abstract by omitting the description "differential coupling" in the context of the analyzed mutants. This expression seems unclear for the reader, who does not know the entire story. The more descriptive way could serve better here.
- Figure 2b requires better text explanations. Also, Figure 2 includes sophisticated biochemical analyses of the complex architecture. The author may consider to include some simple schematic representations (which may be in fact partially overlapping with the final model) that will facilitate the understanding of conclusions by non-specialists.

Reviewer #4 (Remarks to the Author)

This well executed and interesting study investigates the role of Mdm10 in the ER-mitochondria encounter complex (ERMES) and in the Sorting and assembly machinery (SAM) complex. SAM and ERMES complexes have completely different functions, it is thus to date, a puzzle that Mdm10 is part of these two complexes.

The manuscript is divided in three parts. In the first part, the authors describe separation of function alleles of Mdm10 that preferentially interact with SAM and ERMES (Fig 1, 2 and 3).

The second part aims at deciphering the chain of interactions within the ERMES complex, and concludes that the role of Mdm10 within the ERMES complex is merely to anchor the protein to the OMM.

The third part is an electrophysiological study on Mdm10 channel activity, which shows that

Mdm10 forms a conducting pore that is modulated by Tom22.

This manuscript contains highly interesting data but is in clear contradiction with the literature in many cases. It is unclear however which, from the literature and from this paper is correct. The authors should thus state clearly the discrepancy between their data and these of the literature, and why their data and their interpretation is more solid.

1. Starting with the separation of function. The Y296,301A,F298A mutant of Mdm10 shows a growth defect on YPG at 37C. It is claimed that this mutant is solely defective in SAM complex activity, and thus that the thermosensitivity on YPG is due to a SAM defect. A previous paper has achieved separation of function between ERMES-related phenotypes and SAM-related phenotypes of MDM10 deletion by overexpressing MCP1 (Rapoport lab, JCS). This led to a rescue of the ERMES-related phenotypes but not of the SAM-related phenotypes. In this case, growth at 37C on YPG was normal, while TOM complex assembly was defective (indicating that the SAM-related activity of MDM10 was not rescued). Thus the growth defect at 37C on YPG was solely attributed to the ERMES-related function of MDM10. So what does the growth defect of the Y73,75A mutant mean? A straightforward explanation is that the separation of function is not "clean" and that some ERMES-related functions are altered in this mutant. Is there any other explanation? In any case, the fact that separation of function has already been achieved using MCP1 overexpression should be mentioned and this discrepancy explained.

2. Another thing contradicting published literature is the alkaline extraction of Mdm34. Yougman et al. have published the exact opposite (Youngman et al, JCB 2004). In this case, the experimental approach is identical and leading to opposite results, meaning that the difference must be technical. In any case, the discrepancy should be spelled out and the reasons to trust the results in this manuscript, rather than those in the previous one, should be explained.

3. One more thing is the chain of interaction within the ERMES complex. Kornmann et al. have performed a similar analysis using subcellular localization rather than coimmunoprecipitation as a readout for interaction (Kornmann et al. Science 2009). The results are clearly contradictory: They find that Mdm34 localizes to the mitochondria in the absence of Mdm10. In the same conditions, Mmm1 and Mdm12 localize to the ER. Their conclusion is that Mdm12/Mmm1 needs Mdm10 in order to interact with Mdm34. In the present manuscript, Mmm1 interacts with Mdm34 (albeit with reduced intensity) in the absence of Mdm10. How can they do it if they are indeed on different organelles? Does that mean that the complex gets reunified in the test tube after cell lysis and membrane solubilization?

According to Kornmann et al. Mdm10 is not merely "the stable membrane anchor of ERMES at mitochondria" since Mdm34 is, alone, able to localize to mitochondria, but unable to interact with Mdm12 and Mmm1 in the absence of Mdm10.

The discrepancy should be stated and explained.

4. One conclusion of the electrical recordings is that Tom22 reconstituted in the same membrane as Mdm10, influences Mdm10 gating. According to the authors this shows "strong evidence that Mdm10 plays a direct role in the import of Tom22." (the same sentence is repeated in the results and discussion section). How this results leads to this conclusion is not trivial and should be explained.

However the same authors have published that Tom22 and Mdm10 do not interact (Meisinger 2004 Dev Cell). Here is a sentence from this paper : "mdm10Δ mitochondria were still able to import the Tom22 precursor and integrate it into the outer membrane, leading to a comparable fragment formation as in wild-type mitochondria ". Here is another sentence: "Mdm10 is a specific component of the SAM complex and promotes the final steps of assembly of the TOM complex that involves the association of Tom40 with Tom22 and small Tom proteins." Thus, according to this previous paper, it is not the import into the mitochondrial membrane that is affected by Mdm10, but its association with TOM (in any case this paper cannot be referenced to say that "Mitochondria

lacking Mdm10 are impaired in the import of the Tom22 precursor into the mitochondrial outer membrane and the SAM-Mdm10 complex binds the precursor of Tom22 (refs. 18,32)"). In the electrical recording, TOM is absent, thus the effect on Mdm10 are hard to rationalize. It is thus non trivial to say that it shows "strong evidence that Mdm10 plays a direct role in the import of Tom22."

The authors propose their new models without mentioning the discrepancies. These discrepancies should be spelled out and explained in order to avoid confusion in the field. If any of the previous conclusion was wrong, or any technique inadequate, this needs to be made clear to close the debate once and for all.

Additional minor points:

-The sentence "Mdm10 forms a β -barrel channel that mediates protein import" in the abstract is misleading, as it immediately suggests that Mdm10 works as an import channel like Tom40 or Sec61. The data herein indicates that Mdm10 forms a channel (like most β -barrels), and is important for protein import, but not that it works as an import channel.

-Fig 5c: typo Y298A,F298A should read Y296,F298A

-page 11, the sentence: "However, the initial translocation of β -barrel precursors to a protease-protected location was only mildly reduced in Mdm10Y73,75A mitochondria (Fig. 5d), indicating that the subsequent SAM- and Mdm10-dependent steps were mainly impaired in the mutant mitochondria"

may read easier like this:

"However, the initial translocation of β -barrel precursors to a protease-protected location was only mildly reduced in Mdm10Y73,75A mitochondria (Fig. 5d), indicating that it was the subsequent SAM- and Mdm10-dependent steps that were mainly impaired in the mutant mitochondria"

-A better explanation of the electrical readings would be useful. For exemple, the number of gating pores is unclear. How is it calculated (I guess from the mean/variance plot)? What is the evidence that the fusing vesicle contained only one functional unit of the channel? How many independent traces were used to compute the mean/variance plots? How was it computed? By the way, is it mean variance (the variance of the mean) or mean/variance (the mean plotted against the variance)? References could be very useful here.

Reviewer #1:

This is a nice manuscript on a very complex question: how an Eucariotic cell is structurally built up and more specifically how mitochondria is connected with the ER. Here the authors add additional evidence which of a set of four putative proteins is responsible for the interaction. Caused by the lack of high resolution structure the interaction is basically shown by biochemical assays.

Previous publications revealed a set of proteins essential for the function, here the authors could show binding areas and subsequently suggest the strongest interacting proteins. In absence of structural information a combination of methods allows to reveal interaction, In my opinion while waiting for the structure this is the best what one can do.

Knowledge on this interaction is fundamental for the understanding of cellular function.

The experiment are carefully done and described sufficiently.

The manuscript is partly difficult to read and to get the novelty. In the result section previous results are not clearly separated from novel once. The novelty would be more aparent if a number of comments are places into the introduction. Also, in this section I do not expect to review all previous works but rather one reference to an earlier work on which the result is based. Somewhat difficult for outsider to read a number of previous publication to understand the validity of an evaluation of a finding.

As suggested by the reviewer, we have revised the manuscript to differentiate between the previous knowledge and the novel findings presented in the manuscript. We discuss our findings in comparison to the published data about the localization of the ERMES subunits and the molecular organization of the complex. We describe the phenotypes of the *mdm10* deletion strains studied previously and point out that it was unknown which Mdm10 functions were specific for the ERMES- and the SAM-bound forms. We explain the technical differences that led to controversial views on the membrane integration of Mdm34 and describe in detail why Mdm34 is a peripheral membrane protein. The role of SAM-bound Mdm10 in membrane integration of the precursor of Tom22 is now presented by new data (Fig. 5e) and a more detailed explanation.

Reviewer #2

Minor comments

1. The authors propose that Mdm10 shuttles between ERMES and SAM. However, the data are also consistent with another model in which two different populations of Mdm10 stably associate with ERMES or SAM without exchange. Tom7 may help prevent Mdm10 from being recruited exclusively to SAM during the assembly of ERMES.

Following the suggestion of the reviewer, we now discuss the Tom7-controlled segregation of Mdm10 in more detail and cite the relevant literature. We point out that the levels of Tom7 influence the association of Mdm10 with SAM and ERMES. Increased levels of Tom7 lead to a reduction of SAM-Mdm10 and an enrichment of Mdm10 at ERMES. In the absence of Tom7 the SAM-Mdm10 complex accumulates and the recruitment of Mdm10 into ERMES is decreased.

2. The authors reconstituted purified Mdm10 into liposomes. Can the proteoliposomes import Tom22?

Proteoliposomes containing purified Mdm10 specifically react with Tom22 (Fig. 6d), yet binding of Mdm10 to SAM is required for the complete import of Tom22. We include the new Fig. 5e, which shows that membrane-integration of the precursor of Tom22 is impaired in the *mdm10* mutant, which is defective in binding of Mdm10 to the SAM complex, in agreement with the observation that the precursor of Tom22 is associated with the SAM-Mdm10 complex *in organello*. To provide further evidence for a role of Mdm10 in the interaction with Tom22, we show in the new Supplementary Fig. 3b that a purified Tom22 construct, which lacks the cytosolic domain but contains the transmembrane segment, pulls down Mdm10 from lysed mitochondria. Taken together, we conclude that Mdm10 interacts with the precursor of Tom22, but the complete import reaction into the mitochondrial outer membrane is mediated by the SAM-Mdm10 complex.

Reviewer #3:

A few points require experimental clarification and/or discussion

- In Fig. 1 and in many other places either the double mutant in the aromatic string (YF) or the triple mutant (YYF) is used. For example the growth test is performed on the triple mutant (Fig. 1b), and complex analyses for the double mutant (Fig. 1c), and this goes on in the subsequent experiments. Figure 2C may suggest that the double mutant has a milder phenotype in comparison with the more robust triple mutant. This point should be clarified with explicit conclusions. The differences in the interaction between Mmm1 and the Mdm10 mutants (Fig. 2C), especially the double mutant, should be quantified on the basis of multiple experiments and statistical significance of differences should be analyzed.

As suggested by the reviewer, we quantified the interaction between Mmm1 and Mdm10 mutants (new Supplementary Figure 2c). The double mutant of Mdm10 in the aromatic string is significantly impaired in the interaction with Mmm1, however, the triple mutant is considerably more defective. Thus, the double mutant has a milder phenotype in comparison to the triple mutant. Importantly, in the new Fig. 5e, we show that membrane insertion of the precursor of Tom22 is neither affected by the double mutant nor the triple mutant, demonstrating that protein import is not affected by a defective interaction of Mdm10 with ERMES, supporting our conclusion that SAM-bound Mdm10 is responsible for precursor insertion (see also the response to point 2 of reviewer 2).

- It will be useful to clarify, either experimentally or by providing a reference, whether the lack of Tom7 affects the ERMES architecture and function.

We have now included a detailed discussion of the role of Tom7 in shuttling of Mdm10 between SAM and ERMES and the phenotypic consequences and cite the relevant literature. Lack of Tom7 leads to a moderate decrease of the levels of Mdm10-ERMES, however, the phospholipid profiles are not affected and the architecture of the ERMES complex is not disturbed. Taken together, we discuss that Tom7 behaves as a non-essential regulatory subunit of ERMES.

- Figure 5 is a little bit confusing. I agree with the authors' conclusions however I feel that the data presentation can be improved. The moderate decrease in the TOM complex correlates well the import efficiency of the mitochondria from the Y73,75A mutant. Thus, the defect in the translocation more or less reflects a mild problem with the TOM translocase. However, the assembly defect of the Tom40 substrate is much bigger, justifying in full the authors' conclusion on the importance of this Mdm10 site for SAM contacts. I find the control imports of presequence precursors and the entire discussion on the stoichiometry of TOM and TOM complexes dispensable. Negative controls are in general very useful, but Fig. 5e can be easily moved to the Supplement here following shortening of the text on this aspect.

We moved Fig. 5e into the Supplement and removed the discussion on the stoichiometry of protein translocases as suggested.

- The interesting electrophysiology experiments should be complemented by the analysis of Tom22 import into the mutants of Mdm10 that affect ERMES, SAM and Tom7 interactions.

We added the new Fig. 5e that shows the import of Tom22 in the different mutants of Mdm10 that affect ERMES, SAM and Tom7 interactions. Our analysis reveals that the import of Tom22 is specifically impaired in the Mdm10 mutant, which is defective in binding to the SAM complex, fully supporting our conclusions.

Further points:

- I suggest considering a slight change in the abstract by omitting the description "differential coupling" in the context of the analyzed mutants. This expression seems unclear for the reader, who does not know the entire story. The more descriptive way could serve better here.

We have modified the abstract.

- Figure 2b requires better text explanations. Also, Figure 2 includes sophisticated biochemical analyses of the complex architecture. The author may consider to include some simple schematic representations (which may be in fact partially overlapping with the final model) that will facilitate the understanding of conclusions by non-specialists.

We now provide a detailed description of Fig. 2b. In addition, we show a model in the new Fig. 2d that summarizes the findings of Fig. 2, i.e. that one side of the Mdm10 beta-barrel interacts with the SAM complex, while the other side associates with ERMES.

Reviewer #4:

1. Starting with the separation of function. The Y296,301A,F298A mutant of Mdm10 shows a growth defect on YPG at 37C. It is claimed that this mutant is solely defective in SAM complex activity, and thus that the thermosensitivity on YPG is due to a SAM defect. A previous paper has achieved separation of function between ERMES-related phenotypes and SAM-related phenotypes of MDM10 deletion by overexpressing MCP1 (Rapoport lab, JCS). This led to a rescue of the ERMES-related phenotypes but not of the SAM-related phenotypes. In this case, growth at 37C on YPG was normal, while TOM complex assembly was defective (indicating that the SAM-related activity of MDM10 was not rescued). Thus the growth defect at 37C on YPG was solely attributed to the ERMES-related function of MDM10. So what does the growth defect of the Y73,75A mutant mean? A straightforward explanation is that the separation of function is not "clean" and that some ERMES-related functions are altered in this mutant. Is there any other explanation? In any case, the fact that separation of function has already been achieved using MCP1 overexpression should be mentioned and this discrepancy explained.

As suggested by the reviewer, we now discuss the findings of the Rapoport lab (Tan et al., 2013 JCS) in more detail and describe that overexpression of Mcp1 partially restores some *mdm10* defects, but does not lead to a functional separation of the different Mdm10 populations. Overexpression of Mcp1 partially rescues the growth of *mdm10* deletion strains at lower temperatures, but fails to restore growth at 37°C on YPG (Tan et al., 2013, Fig. S1A). The mitochondrial tubular network is restored, but the levels of cardiolipin are not restored by the

overexpression of Mcp1. Thus, the ERMES-related functions of Mdm10 are only partially restored by overexpression of Mcp1. The biogenesis of the TOM complex was still impaired in the Mcp1 overexpression mutant (Tan et al., 2013, Fig. 5C) and likely contributes to the growth defect at 37°C on YPG. Thus, a clear separation of ERMES and SAM-related functions of Mdm10 was not possible by overexpression of Mcp1. Similarly, overexpression of Mcp2 (Tan et al., 2013) and Mdm31 (Tamura et al., 2012) only partially rescued growth and mitochondrial morphology of *mdm10* deletion strains. Our point mutants of Mdm10 thus represent the first tool to selectively separate the functions of Mdm10 at ERMES and SAM.

2. Another thing contradicting published literature is the alkaline extraction of Mdm34. Youngman et al. have published the exact opposite (Youngman et al, JCB 2004). In this case, the experimental approach is identical and leading to opposite results, meaning that the difference must be technical. In any case, the discrepancy should be spelled out and the reasons to trust the results in this manuscript, rather than those in the previous one, should be explained.

We now explain the technical differences of the experimental approaches used by us and Youngman et al. (2004) in detail. Youngman et al. performed the alkaline extraction under milder conditions (pH 11.0), not at the standard condition of pH 11.5. We performed alkaline extraction at pH 11.5 and clearly show a membrane extraction of Mdm34 and Mdm12 (Fig. 4b). Under milder conditions Mdm12 and most of Mdm34 remain in the membrane sheets, although even at pH 10.8 a fraction of Mdm34 is extracted (Fig. 4b). In contrast, Mdm10, Mmm1 and various membrane-integrated control proteins remain fully in the membrane sheets at both conditions (pH 11.5 and pH 10.8). Together with the lack of any detectable transmembrane segment in Mdm34 (analyzed with five different prediction programs), these findings demonstrate that Mdm34 behaves as a peripheral membrane protein.

3. One more thing is the chain of interaction within the ERMES complex. Kornmann et al. have performed a similar analysis using subcellular localization rather than coimmunoprecipitation as a readout for interaction (kornmann et al science 2009). The results are clearly contradictory: They find that Mdm34 localizes to the mitochondria in the absence of Mdm10. In the same conditions, Mmm1 and Mdm12 localize to the ER. Their conclusion is that Mdm12/Mmm1 needs Mdm10 in order to interact with Mdm34. In the present manuscript, Mmm1 interacts with Mdm34 (albeit with reduced intensity) in the absence of Mdm10. How can they do it if they are indeed on different organelles? Does that mean that the complex gets reunified in the test tube after cell lysis and membrane solubilization?

According to Kornmann et al. Mdm10 is not merely "the stable membrane anchor of ERMES at mitochondria" since Mdm34 is, alone, able to localize to mitochondria, but unable to interact with Mdm12 and Mmm1 in the absence of Mdm10.

The discrepancy should be stated and explained.

As suggested by the reviewer, we now discuss the findings by Kornmann et al. (2009 Science) and our findings in more detail. We point out that Kornmann et al. reported that a GFP-tagged Mdm34 was still found in association with mitochondria in the absence of Mdm10, however, it was not determined if this Mdm34 was stably anchored in the mitochondrial outer membrane or just peripherally attached. We have softened our conclusion on Mdm10 and now state that it forms a stable membrane anchor of ERMES (and not as in the first version of the manuscript "Mdm10 forms the only stable membrane anchor"). We also added that Suresh et al. (2015 MBoC) observed that upon starvation the localization of a large fraction of GFP-tagged Mdm34 shifted from ERMES foci to the cytosol in a reversible manner, supporting our conclusion that Mdm34 is only peripherally attached to the mitochondrial outer membrane. Importantly, Kornmann et al. also showed that an artificial tethering construct (ChiMERA), which connects ER and mitochondria, restores the mitochondrial morphology of cells lacking Mdm12 or Mdm34, but not of cells lacking

Mmm1 or Mdm10. Thus, the artificial tether can replace functions of the two peripheral membrane proteins Mdm12 and Mdm34, but not that of the integral membrane proteins Mmm1 and Mdm10 in line with our conclusion that Mdm12 and Mdm34 form the bridge between Mmm1 and Mdm10. We now point out that Mmm1 and Mdm10 likely perform additional functions that cannot be substituted for by an artificial tether.

In their model ofERMES-mediated ER-mitochondria tethering, Kornmann et al. (2009, Fig. 3C legend) state: "Mmm1 interacts with Mdm10, a OMM beta-barrel protein. Mdm34 and Mdm12 promote this association, most probably via direct association." Our biochemical findings agree with this model and directly demonstrate that the association of Mmm1 with Mdm10 requires Mdm12 and Mdm34.

4. One conclusion of the electrical recordings is that Tom22 reconstituted in the same membrane as Mdm10, influences Mdm10 gating. According to the authors this shows "strong evidence that Mdm10 plays a direct role in the import of Tom22." (the same sentence is repeated in the results and discussion section). How this results leads to this conclusion is not trivial and should be explained.

However the same authors have published that Tom22 and Mdm10 do not interact (Meisinger 2004 Dev Cell). Here is a sentence from this paper : "mdm10Δ mitochondria were still able to import the Tom22 precursor and integrate it into the outer membrane, leading to a comparable fragment formation as in wild-type mitochondria ". Here is another sentence: "Mdm10 is a specific component of the SAM complex and promotes the final steps of assembly of the TOM complex that involves the association of Tom40 with Tom22 and small Tom proteins." Thus, according to this previous paper, it is not the import into the mitochondrial membrane that is affected by Mdm10, but its association with TOM (in any case this paper cannot be referenced to say that "Mitochondria lacking Mdm10 are impaired in the import of the Tom22 precursor into the mitochondrial outer membrane and the SAM-Mdm10 complex binds the precursor of Tom22 (refs. 18,32)").

In the electrical recording, TOM is absent, thus the effect on Mdm10 are hard to rationalize. It is thus non trivial to say that it shows "strong evidence that Mdm10 plays a direct role in the import of Tom22."

The authors propose their new models without mentioning the discrepancies. These discrepancies should be spelled out and explained in order to avoid confusion in the field. If any of the previous conclusion was wrong, or any technique inadequate, this needs to be made clear to close the debate once and for all.

We agree with the reviewer that Meisinger et al. (2004) is not a suitable citation for "Mitochondria lacking Mdm10 are impaired in the import of the Tom22 precursor into the mitochondrial outer membrane and the SAM-Mdm10 complex binds the precursor of Tom22" and have modified the manuscript accordingly. Meisinger et al. (2004) was the first paper describing the association of Mdm10 with the SAM complex; there it was concluded that Mdm10 plays a role in the assembly of the TOM complex. The subsequent analysis and in particular the new data in Fig. 5e demonstrate that SAM-bound Mdm10 is critical for the actual membrane insertion of Tom22. In Fig. 5e the role of Mdm10 in membrane insertion of the precursor of Tom22 is analyzed by generation of a specific Tom22 proteolytic fragment, which is characteristic for the membrane-inserted form of Tom22. Using this assay, we studied the import of Tom22 in the different mutants of Mdm10 that affect ERMES, SAM and Tom7 interactions. Our analysis reveals that the import of Tom22 is specifically impaired in the Mdm10 mutant, which is defective in binding to the SAM complex, fully supporting our conclusions. The import defect leads to defective assembly of the TOM complex (Fig. 5b), explaining the findings reported in Meisinger et al. (2004). To provide further evidence for a role of Mdm10 in the biogenesis of Tom22, we show in the new Supplementary Fig. 3b that a purified Tom22 construct, which lacks the cytosolic domain but contains the transmembrane segment, pulls down Mdm10 from lysed mitochondria.

Taken together with the stimulation of the channel activity of reconstituted Mdm10 by Tom22 (Fig. 6d,f) and the binding of the Tom22 precursor to the SAM-Mdm10 complex *in organello* (but not to the SAM_{core} complex; references cited in the revised manuscript), we conclude that Mdm10 interacts with the precursor of Tom22, but the complete import reaction into the mitochondrial outer membrane is mediated by the SAM-Mdm10 complex.

Additional minor points:

-The sentence "Mdm10 forms a β -barrel channel that mediates protein import" in the abstract is misleading, as it immediately suggests that Mdm10 works as an import channel like Tom40 or Sec61. The data herein indicates that Mdm10 forms a channel (like most β -barrels), and is important for protein import, but not that it works as an import channel.

We have modified the abstract as suggested.

-Fig 5c: typo Y298A,F298A should read Y296,F298A

We have corrected the labeling of Fig. 5c.

-page 11, the sentence: "However, the initial translocation of β -barrel precursors to a protease-protected location was only mildly reduced in Mdm10Y73,75A mitochondria (Fig. 5d), indicating that the subsequent SAM- and Mdm10-dependent steps were mainly impaired in the mutant mitochondria"

may read easier like this:

"However, the initial translocation of β -barrel precursors to a protease-protected location was only mildly reduced in Mdm10Y73,75A mitochondria (Fig. 5d), indicating that it was the subsequent SAM- and Mdm10-dependent steps that were mainly impaired in the mutant mitochondria"

We have changed the sentence as proposed by the reviewer.

-A better explanation of the electrical readings would be useful. For example, the number of gating pores is unclear. How is it calculated (I guess from the mean/variance plot)? What is the evidence that the fusing vesicle contained only one functional unit of the channel? How many independent traces were used to compute the mean/variance plots? How was it computed? By the way, is it mean variance (the variance of the mean) or mean/variance (the mean plotted against the variance)? References could be very useful here.

Following the suggestion of the reviewer, we included a more detailed description of the electrophysiological measurements in the Methods section and cite relevant references. We have checked that the correct term mean-variance plot is used throughout the revised manuscript (the mean current plotted against the current variance) and added a reference to describe how the mean-variance plots were generated (Patlak, 1993; ref. 61). We used mean-variance plots to determine the number of gating pores. Each mean-variance plot is derived from an individual current recording at a given voltage applied. We show one representative for at least five current recordings at a specific voltage. These data sets were obtained from more than five different reconstitutions using three independent preparations of Mdm10.

Reviewer #4 (Remarks to the Author)

Ellenrieder et al. have now more thoroughly discussed some of the discrepancies (and even experimentally addressed some of them).

Some of them however remain and should be addressed at least textually.

1.-the "cleanliness" of the separation alleles and whether the SAM-associated function of Mdm10 contributes to the growth defect.

The authors state that Mcp1 overexpression cannot restore growth to full level and especially on YPG at 37C, citing figure S1 of Tan et al. 2013. I would like to direct them to the Figure 1 of the same paper, showing a full rescue at 37C on YPG. The difference stems likely from the different expression systems: in figure 1 Mcp1 is expressed from its endogenous promoter on a 2micron plasmid, while in figure S1, it is expressed from a CEN/ARS plasmid from the TPI promoter, and additionally bears a C-terminal HA tag. Thus I'd recommend prudence when stating that "...cell growth at higher temperature were not rescued, indicating that crucial functions of Mdm10 were not suppressed by Mcp1".

Mcp1 overexpression rescues almost all lipid defects of ERMES mutants (except cardiolipin), but does not rescue TOM complex assembly. Thus it appears that one can grow fine with improper TOM complex assembly.

This raises the possibility that the growth defect of the Y73,75A mutants is not due to a defect in the SAM-related function of the protein but to a more general crippling of the protein that affects also its other functions. A word of caution is necessary here to interpret the phenotype of these mutants.

2.-the discrepancy with Kornmann et al. 2009 is not solved. Figure 4A claims to show a new model of ERMES organisation. While the non-transmembrane nature of Mdm34 is nicely supported by the carbonate extraction data, the order of interaction (Mdm10-Mdm34-Mdm12-Mmm1) and the role of Mdm10 as MOM anchor of the complex is still not sufficiently supported.

Again, according to Kornmann et al, Mdm10 deletion does not affect the association of Mdm34 to the mitochondria, but instead its association to the Mmm1-Mdm12 subcomplex (- in Kornmann et al. they do not colocalize anymore. - In the present study Mdm34 pull-down with Mmm1 is weaker in the absence of Mdm10).

In that context, the sentence

"Kornmann et al.5 reported that a GFP-tagged Mdm34 was still found in association with mitochondria in the absence of Mdm10, however, it was not determined if this Mdm34 was stably anchored in the mitochondrial outer membrane or just peripherally attached"

is beside the point.

Mdm10 absence does not cause problems in Mdm34 attachment to mitochondria, but causes problems in Mdm34 attachment to the rest of the complex. The stability of Mdm34 attachment to mitochondria is thus irrelevant here. By the way, peripheral does not equate unstable, transmembrane does not equate stable and sentences such as "Mdm10 is the ERMES core component that is stably anchored in the lipid phase of the mitochondrial outer membrane." should be changed by something like "Mdm10 is the sole MOM transmembrane protein of the ERMES complex".

A model that satisfies past and present data looks like the current model without a transmembrane domain in Mdm34.

If the authors want to promote their new model, they need to provide evidence that Mdm10 does not participate in the interaction of Mdm34 with Mmm1-Mdm12. Mdm34 and Mmm1-Mdm12

should be appropriately colocalized in the absence of Mdm10, and the remaining interaction between Mmm1 and Mdm34 observed herein in the absence of Mdm10 should be shown to be robust and no artifact generated by the reunion of Mdm34 with the remaining complex after cell lysis.

Here is an experiment that could test the latter: mix equal amounts of cells expressing MMM1-protA and Mdm34GFP (or HA, anything that changes the electrophoretic mobility of the protein), with cells expressing untagged Mmm1 and Mdm34; and do it in otherwise wildtype or *mdm10*-delete cells. Perform the pull down and blot with anti-Mdm34 antibody (which recognises both normal and tagged Mdm34). If the interaction is genuine, then Mmm1-protA should pull down only GFP-tagged Mdm34. If the interaction happens after lysis, then Mmm1 should pull down equal amounts of tagged and untagged Mdm34.

In the absence of any new data, the authors should substantially soften their stance on their new model (remove fig. 4A) and reconsider the role of Mdm10 as sole stable anchor for the ERMES complex on the MOM. Again the stability of the association of ERMES to mitochondria is not the issue here. The problem appears to be the association of the complex itself.

Reviewer #4:

Ellenrieder et al. have now more thoroughly discussed some of the discrepancies (and even experimentally addressed some of them).

Some of them however remain and should be addressed at least textually.

1.-the "cleanliness" of the separation alleles and whether the SAM-associated function of Mdm10 contributes to the growth defect.

The authors state that Mcp1 overexpression cannot restore growth to full level and especially on YPG at 37C, citing figure S1 of Tan et al. 2013. I would like to direct them to the Figure 1 of the same paper, showing a full rescue at 37C on YPG. The difference stems likely from the different expression systems: in figure 1 Mcp1 is expressed from its endogenous promoter on a 2micron plasmid, while in figure S1, it is expressed from a CEN/ARS plasmid from the TPI promoter, and additionally bears a C-terminal HA tag. Thus I'd recommend prudence when stating that "...cell growth at higher temperature were not rescued, indicating that crucial functions of Mdm10 were not suppressed by Mcp1".

Mcp1 overexpression rescues almost all lipid defects of ERMES mutants (except cardiolipin), but does not rescue TOM complex assembly. Thus it appears that one can grow fine with improper TOM complex assembly.

We carefully studied the findings of the publication by Tan et al. (2013 J. Cell Sci.). The authors performed a genomic screen to identify novel high-copy suppressors of the *mdm10* deletion strain. They introduced the 2 μ plasmid pFL44 encoding for genomic fragments into an *mdm10* deletion strain and analyzed the growth at 30°C and 37°C on non-fermentable carbon source. In this screen they identified a plasmid encoding *ARS1523*, *YOR228C (MCP1)* **and** *WTM2*. This plasmid suppressed the growth defect of the *mdm10* deletion strain at 37°C (Tan et al., 2013; Fig. 1A), but was not used for further biochemical and phenotypic studies. Thus, *WTM2* encoding a transcriptional modulator and the autonomously replicating sequence *ARS1523* were **not** analyzed in this study.

Instead, to directly analyze the role of Mcp1, the ORF of Mcp1 alone was introduced into a pYX142 plasmid under the control of a TPI promoter. Overexpression of Mcp1 did not rescue the growth at 37°C on YPG (Tan et al., 2013; Fig. S1A). Importantly, Tan et al. conclude in their paper that overexpression of Mcp1 failed to rescue the *mdm10* deletion growth defect at high temperature and non-fermentable carbon source (37°C and YPG medium; Fig. S1A). (Tan et al., 2013, page 3564). Thus, Tan et al. did not observe a full rescue of the growth of the *mdm10* deletion strain by overexpression of Mcp1 at 37°C on YPG in agreement with the description in our manuscript.

For biochemical and phenotypic characterization, Mcp1 was overexpressed without an HA-tag, although the HA-tag did not impact the suppression phenotype at lower temperature (Tan et al., 2013; Fig. S5). Biochemical analysis revealed that Mcp1 overexpression did not restore TOM assembly and only partially rescued the cardiolipin levels, indicating that both functions of Mdm10 are required for optimal growth of yeast cells at higher temperature. We describe these findings in our manuscript: "Tan *et al.*²³ showed that overexpression of the mitochondrial outer membrane protein Mcp1 restored the mitochondrial morphology of cells lacking Mdm10, however, cardiolipin levels, protein import into mitochondria and cell growth at higher temperature were not fully rescued, indicating that crucial functions of Mdm10 were not suppressed by Mcp1."

This raises the possibility that the growth defect of the Y73,75A mutants is not due to a defect in the SAM-related function of the protein but to a more general crippling of the protein that affects also its other functions. A word of caution is necessary here to interpret the phenotype of these mutants.

The findings of Tan et al. support our observation that the defect in TOM assembly leads to a growth defect of the Mdm10 Y73,75A mutant at 37°C on YPG. In the Mdm10 Y73,75A strain the cardiolipin levels were not reduced and the ERMES-related defects in mitochondrial morphology were not observed. We show that binding of the mutant form of Mdm10 to the SAM complex is impaired, whereas the association with ERMES remains unaffected. Our study reveals that binding of Mdm10 to SAM promotes the biogenesis of the TOM complex. Several previous studies have shown that the proper assembly of the protein import machinery TOM is important for yeast growth. Yeast mutants defective in the biogenesis of the TOM complex reveal growth defects, in particular on non-fermentable carbon sources at 37°C (e.g. Ishikawa et al., J. Cell Biol. 2004; Waizenegger et al., EMBO Rep. 2005; Dukanovic et al., Mol. Cell. Biol. 2009; Becker et al., Mol. Biol. Cell 2010). We conclude that a disturbed binding of Mdm10 to SAM impairs the formation of the TOM complex and leads to a growth defect of the Mdm10 Y73,75A strain at 37°C on YPG.

Our conclusion is thus in full agreement with the studies of the protein import field and the data obtained by Mcp1 overexpression by Tan and colleagues.

2.-the discrepancy with Kornmann et al. 2009 is not solved. Figure 4A claims to show a new model of ERMES organisation. While the non-transmembrane nature of Mdm34 is nicely supported by the carbonate extraction data, the order of interaction (Mdm10-Mdm34-Mdm12-Mmm1) and the role of Mdm10 as MOM anchor of the complex is still not sufficiently supported.

Again, according to Kornmann et al, Mdm10 deletion does not affect the association of Mdm34 to the mitochondria, but instead its association to the Mmm1-Mdm12 subcomplex (- in Kornmann et al. they do not colocalize anymore anymore. - In the present study Mdm34 pull-down with Mmm1 is weaker in the absence of Mdm10).

In that context, the sentence

"Kornmann et al.5 reported that a GFP-tagged Mdm34 was still found in association with mitochondria in the absence of Mdm10, however, it was not determined if this Mdm34 was stably anchored in the mitochondrial outer membrane or just peripherally attached" is beside the point.

Mdm10 absence does not cause problems in Mdm34 attachment to mitochondria, but causes problems in Mdm34 attachment to the rest of the complex. The stability of Mdm34 attachment to mitochondria is thus irrelevant here. By the way, peripheral does not equate unstable, transmembrane does not equate stable and sentences such as "Mdm10 is the ERMES core component that is stably anchored in the lipid phase of the mitochondrial outer membrane." should be changed by something like "Mdm10 is the sole MOM transmembrane protein of the ERMES complex".

A model that satisfies past and present data looks like the current model without a transmembrane domain in Mdm34.

If the authors want to promote their new model, they need to provide evidence that Mdm10 does not participate in the interaction of Mdm34 with Mmm1-Mdm12. Mdm34 and Mmm1-Mdm12 should be appropriately colocalized in the absence of Mdm10, and the remaining interaction between Mmm1 and Mdm34 observed herein in the absence of Mdm10 should be shown to be robust and no artifact generated by the reunion of Mdm34 with the remaining

complex after cell lysis.

Here is an experiment that could test the latter: mix equal amounts of cells expressing MMM1-protA and Mdm34GFP (or HA, anything that changes the electrophoretic mobility of the protein), with cells expressing untagged Mmm1 and Mdm34; and do it in otherwise wildtype or mdm10-delete cells. Perform the pull down and blot with anti-Mdm34 antibody (which recognises both normal and tagged Mdm34). If the interaction is genuine, then Mmm1-protA should pull down only GFP-tagged Mdm34. If the interaction happens after lysis, then Mmm1 should pull down equal amounts of tagged and untagged Mdm34.

In the absence of any new data, the authors should substantially soften their stance on their new model (remove fig. 4A) and reconsider the role of Mdm10 as sole stable anchor for the ERMES complex on the MOM. Again the stability of the association of ERMES to mitochondria is not the issue here. The problem appears to be the association of the complex itself.

Following the suggestions of the reviewer, we provide additional experimental evidence to exclude a post-lysis artifact of Mdm34 interaction with Mmm1-Mdm12 and have toned down the conclusions on the role of Mdm10 in ERMES by removing Fig. 4a and describing more exactly that Mdm10 is an integral membrane protein in contrast to Mdm34.

In detail, in the new Supplementary Fig. 2d we demonstrate that Mdm34 does not associate with tagged Mmm1 after cell lysis, excluding an unspecific post-lysis interaction.

We report in the manuscript that deletion of *MDM34* inhibited the co-purification of Mdm10 with tagged Mmm1, whereas deletion of *MDM10* attenuated but did not block the co-purification of Mdm34 with Mmm1. We conclude that Mdm10 supports the association of Mdm34 with Mdm12-Mmm1, but is not strictly required for this interaction.

We removed the model Fig. 4a and now describe more clearly that Mdm10 is the integral membrane anchor of the ERMES core components and that Mdm34 does not contain a transmembrane anchor.

We removed the sentence "*Kornmann et al.*⁵ reported that a GFP-tagged Mdm34 was still found in association with mitochondria in the absence of Mdm10, however, it was not determined if this Mdm34 was stably anchored in the mitochondrial outer membrane or just peripherally attached" as requested by the reviewer.

Reviewer #4 (Remarks to the Author)

The authors have addressed my concerns.

In particular, they have performed purifications of ERMES in a setup that allows mixing of components during lysis and showed that post-lysis mixing of ERMES components does not happen in their experimental conditions. They haven't shown that "free" Mdm34 (i.e. from a *mdm10* delete strain) cannot associate with "free" *mmm1-mdm10* subcomplex, but their results above make this possibility unlikely. The discrepancy with Kornmann et al. thus remains unexplained, but now stands beyond the scope of the study.

My conclusions from the Tan et al. 2013 paper are different from that these of the authors, but resolving these issues would require work far beyond the scope of the paper.

Reviewer #4 (Remarks to the Author):

The authors have addressed my concerns.

In particular, they have performed purifications of ERMES in a setup that allows mixing of components during lysis and showed that post-lysis mixing of ERMES components does not happen in their experimental conditions. They haven't shown that "free" Mdm34 (i.e. from a mdm10 delete strain) cannot associate with "free" mmm1-mdm10 subcomplex, but their results above make this possibility unlikely. The discrepancy with Kornmann et al. thus remains unexplained, but now stands beyond the scope of the study.

My conclusions from the Tan et al. 2013 paper are different from that these of the authors, but resolving these issues would require work far beyond the scope of the paper.

We thank all reviewers for their supportive comments.

We agree with reviewer 4 that our experiments show that unspecific post-lysis mixing of ERMES subunits does not occur and that the clarification of the discrepancy to Kornmann and colleagues stands beyond the scope of the manuscript. We followed the conclusions drawn by Tan and colleagues in their publication from 2013, which differ from the reviewer's conclusion.